# How to Price Data: A Market Equilibrium Based Approach

**Pooja Kulkarni** [* 1]  **Parnian Shahkar** [* 2]  **Ruta Mehta** [3]

## Abstract

High-quality data is a key input to modern machine learning models, leading to the emergence of platforms that facilitate the buying and selling of data. A central challenge in these platforms is how the data is priced to balance the interests of both buyers and sellers. Traditional market equilibrium notions, where demand meets supply are commonly used to price goods but do not extend naturally to data due to its non-rivalrous nature, whereby multiple buyers can simultaneously benefit from the same dataset. We therefore introduce a new notion of equilibrium for data pricing based on Nash equilibrium and study it in settings where data may be complementary or substitutable, focusing on the canonical utility models for each, namely Leontief and linear, respectively. We show that equilibrium prices fail to exist for linear utilities even with homogeneous buyers and two sellers, while establishing strong existence, efficiency, and polynomial-time computation guarantees for Leontief utilities in general markets with $n$ homogeneous buyers and $m$ sellers. We further examine the role of platform mediation and price discrimination in enabling *optimal* equilibrium outcomes efficiently. On the technical front, we develop a novel proof technique based on systematically reducing the space of candidate equilibria through the *graph-of-deviations*, which may be of independent interest.

## 1. Introduction

As machine learning models increasingly permeate society, most notably with the rise of generative AI and large language models, data has emerged as a critical economic asset, serving as the primary input that fuels these systems. Naturally, a number of marketplaces have been established to facilitate the buying and selling of data (Bergemann et al., 2018; Agarwal et al., 2019; Pei, 2020; Zhang et al., 2023; Bonatti et al., 2024) resulting in the need for principled systems for pricing and trading data.

A typical market consists of sellers with goods to sell, and buyers who want to buy these goods using their monetary budget. Markets for traditional assets have been extensively studied within economics and computer science (Arrow & Debreu, 1954; Roughgarden, 2010), and *market equilibrium* pricing is considered *the* central solution concept due to its remarkable properties, including stability, welfare, revenue and fairness (Orlin, 2010; Jain & Vazirani, 2010b) guarantees. This naturally motivates pricing and trading data through market equilibrium within data marketplaces.

However, unlike traditional assets such as land or labor, data is *non-rivalrous* and *freely replicable*: the same dataset can simultaneously benefit multiple agents, and additional copies can be distributed at near-zero marginal cost. This fundamental distinction alters how markets with data operate. To illustrate, consider the classical notion of competitive equilibrium (CE), where prices equate supply and demand. Suppose a seller owns a divisible, rivalrous good such as a piece of land, and there are two buyers with budgets \$1 and \$2. At CE, the land must be fully purchased and budgets fully spent, resulting in allocations of ¹/₃ and ²/₃ of the land to the two buyers at a total price of \$3. In contrast, if the seller offers a dataset, free replicability allows simultaneous supply to multiple buyers. Pricing the dataset at \$2 or \$3, for instance, yields the same revenue but with widely different allocations, namely full copy to each buyer at price \$2, and fractions similar to the land example at price \$3. Since there is no natural notion of supply i.e., whether to match total demand to one unit, give each buyer a full copy, or something else entirely, classical competitive equilibrium is ill-suited for pricing data. This leads to the central question of this paper:

---
[*]Equal contribution  [1]Data Science Institute, University of Chicago, USA [2]Department of Computer Science, University of California at Irvine, USA [3]Department of Computer Science, University of Illinois at Urbana-Champaign, USA. Correspondence to: Pooja Kulkarni <kulkarnipooja96@gmail.com>, Parnian Shahkar <shahkarp@uci.edu>.

*Proceedings of the 43ʳᵈ International Conference on Machine Learning*, Seoul, South Korea. PMLR 306, 2026. Copyright 2026 by the author(s).

> *Given a data market, is there a notion of market equilibrium under which properties such as stability, welfare, and revenue guarantees, analogous to those in classical markets, can be preserved?*

In this paper, we answer this question by (i) proposing a new notion of equilibrium tailored to data markets, and (ii) analyzing this notion in two canonical data markets, establishing results on existence, polynomial-time computability, and welfare and revenue guarantees.

### 1.1. Model and Contributions

If we take a step back, equilibrium prices in traditional markets are fundamentally stable: no seller can increase revenue by changing their price, and at the current prices, no buyer can increase value by changing their demand. Our equilibrium notion aims to achieve this fundamental desideratum.

Formally, we consider a market with $m$ sellers and $n$ buyers. Each seller $j \in [m]$ has (without loss of generality) one unit of dataset $j$ to sell. Each buyer $i \in [n]$ has a budget $b_i$ to buy these datasets. For each buyer–seller pair $(i, j)$, let $x_{ij} \in [0, 1]$ denote the fraction of seller $j$'s dataset bought by buyer $i$. Buyer $i$'s preferences are captured by a valuation function $u_i : (x_{ij})_{j \in [m]} \to \mathbb{R}$, which specifies the utility derived from the acquired datasets. All our results extend to integral datamarkets, i.e., $x_{ij} \in \{0, 1\}$ for all buyer-seller pairs.

At given prices $\mathbf{p} = (p_j)_{j \in [m]}$, buyer $i$ can demand any bundle $\mathbf{x}_i \in [0, 1]^m$ ($\mathbf{x}_i \in \{0, 1\}^m$ for the integral datamarkets) of datasets, that she can afford, that is $\sum_{j \in [m]} x_{ij} p_j \leq b_i$. Clearly, she will demand the one that maximizes her value $u_i(\mathbf{x}_i)$, *i.e.,* her *optimal bundle*. Thus, the induced demand vector $\mathbf{x}_i$ is essentially a function of prices $\mathbf{p}$ of the datasets. The resulting revenue of seller $j$ is the total demand of her dataset $j$ times its price, $(p_j * \sum_{i \in [n]} x_{ij})$.

**Equilibrium.** Now, it is natural to define equilibrium as a stable state of this pricing game, aka *Nash equilibrium*. That is, prices $\mathbf{p}^*$ at which every buyer buys/demands her optimal bundle, and no seller $j$ can earn higher revenue by changing the price $p_j^*$ of her dataset given that the buyers' demand will change accordingly.

The structure of equilibrium outcomes, including its existence, depends critically on how buyers derive value from multiple datasets. In particular, the datasets can be *substitutable* or *complementary*; see the following scenarios.

*Complementary Data.* Suppose several pharmaceutical companies aim to develop a vaccine and test its efficacy across the population. Thus, they need patient-level data from hospitals from various geographic regions since they hold data correlated with the local demographic population. To accurately evaluate vaccine efficacy across the population, the companies require data from *all* demographic locations in proportions that represent the overall population. In this setting—with hospitals as data sellers and pharmaceutical companies as buyers—the joint access to multiple datasets yields much greater value to the buyers than individual datasets. Such interdependence in buyers' preferences are known as *complementarity*.

*Substitute Data.* Suppose a large, unlabeled public dataset exists and multiple individual sellers independently label different portions of it. For example, in building a language model, different contributors may label text or translate sentences in parallel. Each labeled dataset contributes independently to improving the model's performance: the more labeled data a buyer obtains, the better the downstream predictions. In such cases, access to multiple datasets provides value no greater than the sum of their individual contributions, reflecting the independent value contributed by each seller. Substitute functions of this form are also commonly assumed in federated learning and crowdsourced data markets (Henighan et al., 2020; Murhekar et al., 2025).

**Utility model.** In this work, we address both complementary and substitutable data settings under the canonical and well-studied valuation functions of *Leontief* and *linear* respectively. See Section 2 for formal definitions.

Finally, we assume that buyers are homogeneous in the sense that they compete on the same downstream task. For example, in the pharmaceutical example, all buyers evaluate vaccine efficacy for the same target population and thus share the same training and test distributions. Consequently, buyers evaluate datasets in a similar manner and hence have the same valuation function. However, *their budgets are different*.

*Utility Thresholds.* A primary use case of data is training ML models, which typically requires sufficiently large datasets for training to be meaningful. To capture this, we introduce utility thresholds for buyers. Specifically, for each agent $i$, a utility threshold $u_i^{\min} \geq 0$ is specified, meaning that she derives no value from the acquired data bundle $\mathbf{x}_i$ unless $u_i(\mathbf{x}_i) \geq u_i^{\min}$. Here, $u_i^{\min}$ can represent the minimum accuracy required from the trained ML model for it to be meaningful for the downstream task.

The *buyers also differ on their minimum utility thresholds* indicating differences in their downstream algorithms: more efficient algorithms may require less accurate ML-model, corresponding to lower minimum utility requirements.

**Equilibrium Guarantees.** For the above market model and equilibrium notion, we prove the following guarantees:

**Complementary data: Leontief valuation functions.**

- An equilibrium always exists and can be computed in polynomial time.
- Among all equilibria, we can efficiently compute one that simultaneously maximizes buyer welfare, total seller revenue, and fairness across sellers (all sellers receive equal revenue). Moreover, this equilibrium also maximizes the number of buyers who achieve their minimum utility.
- The *best-response-dynamics (BRD)* among buyers and sellers converges to a $(1 + \epsilon)$-approximate equilibrium in polynomial time. This is surprising because convergence of BRD, although sought-after, is rare.
- Experiments suggest that the best-response-dynamics need not converge to the globally optimal equilibrium.

**Substitutable data: Linear valuation functions.**

- In contrast, in this case an equilibrium may fail to exist even with just two sellers.
- An equilibrium does exist with a single buyer implying that platforms can restore equilibrium under linear utilities via price discrimination, i.e., showing different buyers different prices for the same dataset.

**Techniques.** The thresholded nature of our utilities induces discontinuities and non-convexities that rule out standard fixed-point–based existence proofs. We therefore develop a new technique: we partition the strategy space into a polynomial number of *states*, identify a canonical equilibrium candidate in each, and construct a graph of profitable deviations between states. Showing this graph is acyclic implies that a sink corresponds to an equilibrium. We expect this approach to be broadly useful for data markets with non-convex valuations.

**Implications.**

- *Leontief markets:* The platform has the ability to compute equilibrium prices that is simultaneously optimal for buyers and sellers. On the other hand, buyers and sellers can not figure such prices out through an intuitive price-adjustment process like the best-response-dynamics. Thus, platforms *should* mediate pricing, advising sellers on equilibrium prices that ensure fairness and efficiency for all.
- *Linear markets:* Platforms should implement price discrimination to achieve optimal equilibrium, a strategy commonly observed in practice, e.g., airline ticketing or Amazon coupon systems.

*Remark* 1.1. As is common, we assume that the platform helps buyer figure out their valuation function over the datasets while maintaining privacy. This is indeed an important problem that is well-studied in literature. In this paper, we focus on the problem of data pricing.

**Other Related Work.** There is extensive work on data economics, for instance, (Markovich & Yehezkel, 2026) considered platforms where the users are charged for privacy via services. There is also work on equilibrium data pricing. For instance, (Gao et al., 2025) model a data market using a platform that facilitates trade using blockchain technology. They model the market as a two level stackelberg game between three parties – data platform, data sellers and data buyers and investigate equilibrium in this setting. (Jain & Vazirani, 2010a) and (Chaudhury et al., 2026) consider equilibrium pricing in economies with digital goods and data respectively, though their notion of equilibrium differs from ours—they enforce a supply-demand balance by requiring each digital good and dataset respectively to be fully purchased by at least one buyer. Finally, simultaneous work to ours, (Chaudhury et al.) consider data pricing in data markets with the same equilibrium model as ours and give approximate equilibria with non-uniform pricing for substitutable markets. Other prior work has focused on aspects such as privacy, arbitrage-freeness, truthfulness, and revenue maximization. For example, (Zhao et al., 2023) study pricing ML datasets with seller compensation aligned to value contribution; (Lin & Kifer, 2014; Chawla et al., 2019) explore pricing queries over datasets; and (Chen et al., 2019) examine model-based pricing for ML models (instead of data). (Agarwal et al., 2019) consider online feature pricing to maximize total revenue, without considering individual seller incentives. Classical works such as (Admati & Pfleiderer, 1986; 1990) initiate the study of revenue-maximizing strategies for monopolistic data sellers. (Bergemann et al., 2018) study revenue-maximization in single buyer, single-seller market with private information. (Babaioff et al., 2012) study interactive pricing mechanisms in single-buyer, single-seller settings with private information. (Bonatti et al., 2024; Mehta et al., 2021) address multi-buyer scenarios with a single seller, and (Bimpikis et al., 2019; Agarwal et al., 2024) analyze pricing under buyer externalities with monopolistic sellers. For broader surveys, see (Zhang et al., 2023; Pei, 2020; Bergemann & Bonatti, 2019). In Appendix A, we discuss further work related to the closely related notion of Nash Equilibrium, Pricing Games, and markets for rivalrous goods.

## 2. Notation and Preliminaries

We consider a market with $m$ data sellers, $[m] = \{1, 2, \ldots, m\}$ and $n$ buyers, $[n] = \{1, 2, \ldots, n\}$. Buyer $i \in [n]$ has a budget $b_i$. We use $\mathbf{x} = (x_{ij})_{i \in [n], j \in [m]}$ to denote the allocation matrix $- x_{ij}$ is the fraction of seller $j's$ data purchased by buyer $i$. We denote the *per unit* price of seller's datasets with $p_j$ and use $\mathbf{p} = (p_j)_{j \in [m]}$ as the full price vector. Each buyer $i$ has a utility function $u_i : [0, 1]^m \to \mathbb{R}$ that denotes her preference for obtaining a subset of items. Additionally, the minimum required utility of a buyer is represented by $u_i^{min}$. If the buyer cannot

receive a $u_i^{min}$ unit of utility from the market, she does not participate in it. Our results work for both fractional i.e., $x_{ij} \in [0,1]$ and integral $x_{ij} \in \{0,1\}$ markets. The fractional model is interpreted as seller's dataset consisting of i.i.d. samples. Purchasing an $x$-fraction of a dataset corresponds to accessing an arbitrary subsample of that size; by i.i.d.-ness, this preserves the statistical properties of the full dataset. Accordingly, pricing is linear: an $x$-fraction of seller $j$'s data costs $xp_j$. We analyze two valuation function classes: Leontief and linear.

**Leontief Valuations.** Under classic Leontief utilities, each buyer $i$ has a vector $(w_{ij})_{j \in [m]}$. Given an allocation $\mathbf{x}_{i,\cdot} = (x_{ij})_{j \in [m]}$, the utility of $i$ is given by $\min_j \{\frac{x_{ij}}{w_{ij}}\}$. Since the buyers are homogeneous, we drop the subscript $i$ and denote these proportions by $\mathbf{w} = (w_j)_{j \in [m]}$. In our model, each dataset has unit supply, so an agent cannot benefit from purchasing more than one unit of any dataset. We encode this by scaling the proportion vector $\mathbf{w}$ so that $\max_{j \in [m]} w_j = 1$. Then, for any optimal bundle, we have $x_{ij} \leq w_j$ for all $j \in [m]$. Indeed, since $x_{ij} \leq 1$ for all $j$, letting $j^* = \arg\max_j w_j$ gives $x_{ij^*}/w_{j^*} \leq 1$, which upper-bounds $\min_j x_{ij}/w_j$ by 1. Consequently, purchasing more than $w_j$ of dataset $j$ does not increase utility. This normalization is without loss of generality. Finally, to model the fact that buyers might demand a minimum utility $u_i^{min}$ from the market, we convert it to a threshold $\tau_i$. This threshold is interpreted as the buyer requiring to purchase at least $\tau_i w_j$ fraction of sellers $j$'s dataset. Once she does this, the utility of buyer will be $(\min_j \{\frac{\tau_i w_j}{w_j}\}) = \tau_i$ and we can convert any given $u_i^{min}$ into a corresponding $\tau_i$. Therefore, buyer $i$'s utility is

$$u_i(\mathbf{x}) = \begin{cases} 0, & \text{if infeasible,} \\ v_i\left(\min_{j \in [m]} \frac{\min\{x_{ij}, w_j\}}{w_j}\right), & \text{otherwise,} \end{cases}$$ (1)

where infeasible means $\exists j \in [m]$ such that $x_{ij} < \tau_i w_j$. Note that $v_i$ is any strictly monotonically increasing concave function, normalized so that $v_i(0) = 0$. $\tau_i$ is such that $v_i(\tau_i) = u_i^{min}$.

**Linear Valuations.** Under linear utility functions, each buyer $i$ values a full unit of seller $j$'s data at $w_{ij}$. As the buyers are homogeneous, we drop the subscript $i$ and let $\mathbf{w} = (w_j)_{j \in [m]}$ be the common valuation vector across all buyers. Since we assume each seller's dataset consists of i.i.d. samples and fractional sales are implemented by randomly subsampling the data, an $x$-fraction of seller $j$'s dataset provides value $xw_j$ to any buyer. Under linear utilities, buyer $i$'s total value from an allocation is therefore $\sum_{j \in [m]} x_{ij} w_j$.

To capture buyers who require a minimum value to participate, each buyer $i$ specifies a minimum threshold $u_i^{min}$: if the total value she obtains is less than $u_i^{min}$, she leaves the market and purchases nothing. Accordingly, the buyer's utility function can be formalized as

$$u_i(\mathbf{x}) = \begin{cases} 0, & \text{if } \sum_{j \in [m]} x_{ij} v_j < u_i^{min}, \\ v_i\left(\sum_{j \in [m]} x_{ij} w_j\right), & \text{otherwise.} \end{cases}$$ (2)

where $v_i(\cdot)$ can be any (strictly) monotonically increasing and concave function with $v_i(0) = 0$.

**Buyer Welfare.** Given an allocation $\mathbf{x} = (x_{ij})_{i \in [n], j \in [m]}$, the buyer welfare (or simply welfare) is the sum of utilities received by the buyers i.e., $\sum_{i \in [n]} u_i(\mathbf{x})$.

**Seller revenue.** Revenue of seller $j$ is the total money she earns. Therefore, given that the buyers are purchasing as $\mathbf{x} = (x_{ij})_{i \in [n], j \in [m]}$, seller $j$'s total demand is $X_j = \sum_{i \in [n]} x_{ij}$ and her revenue is $p_j X_j$. Note that unlike traditional goods, the same piece of data can be sold to various buyers, therefore $X_j$ can be greater than 1. Given the price vector $\mathbf{p} = (p_j)_{j \in [m]}$, each buyer sets her demand for each dataset to maximize her utility, so $\mathbf{x} = (x_{ij})_{i \in [n] j \in [m]}$ is itself a function of $\mathbf{p}$, and therefore, the revenue of seller $j$ is also a function of $\mathbf{p}$ and we denote it as $R_j(\mathbf{p}) = p_j X_j(\mathbf{p})$. The total revenue of sellers is the sum of seller revenues, i.e. $\sum_{j \in [m]} R_j(\mathbf{p})$.

**Market Equilibrium.** We define a market equilibrium as a set of prices $\mathbf{p} = (p_j)_{j \in [m]}$ and an allocation rule $\mathbf{x} = (x_{ij})_{i \in [n], j \in [m]}$ such that

1. (No seller deviation.) For every seller $j$, $R_j(\mathbf{p}) \geq R_j(p'_j, p_{-j})$ where $p_{-j}$ is the set of prices of all sellers except for $j$, and $p'_j$ is the new price for seller $j$.
2. (No buyer deviation.) For every buyer $i$,

$$u_i((x_{ij})_{j \in [m]}) = \max_{(y_{ij})_{j \in [m]}} \{u_i(y_{ij}) \mid \sum_{j \in [m]} p_j y_{ij} \leq b_i\}.$$

Similarly, we define a $(1 + \epsilon)$ market equilibrium where for every seller, $j$, $(1 + \epsilon)R_j(\mathbf{p}) \geq R_j(p'_j, p_{-j})$ and for every buyer, $i$, $(1 + \epsilon)u_i((x_{ij})_{j \in [m]}) \geq \max_{(y_{ij})_{j \in [m]}} \{u_i(y_{ij}) \mid \sum_{j \in [m]} p_j y_{ij} \leq b_i\}$.

## 3. Complementary Data

In this section, we study the setting in which buyers view the sellers' datasets as perfect complements. Since in our model, a seller's best-response may be non-convex and change discontinuously with others' prices (see Section C.2), Kakutani's theorem does not apply. Therefore, we first establish structural properties that enable us to prove the existence of a Nash equilibrium, and by leveraging these properties, we give an efficient algorithm to compute the *best* Nash

equilibrium—one that is seller-revenue–maximizing, buyer-welfare–maximizing, and fair to every seller. We also show that under best-response dynamics the market converges to a $(1 + \epsilon)$-equilibrium; however, extensive experiments indicate that the resulting equilibrium need not be the best one. This motivates the role of a mediating platform that intervenes to compute the best equilibrium.

*All our results extend to integral markets, where datasets can be sold only in whole units; the proof appears in Appendix E. All omitted proofs from this section are deferred to Appendix C.*

## 3.1. Structural Results

**Definition 3.1** (Active Buyer). A buyer $i$ is *active* if she remains in the market, i.e., she can afford to purchase at least the minimum required amount $\tau_i w_j$ from every seller $j$.

Given the price vector $\mathbf{p} = (p_j)_{j \in [m]}$, an active buyer needs to spend $\tau_i w_j p_j$ to purchase the minimum required amount from each seller $j$. Hence, she is active only if her budget allows her to purchase the minimum amount, i.e. $b_i \geq \tau_i \sum_{j \in [m]} w_j p_j$. Note that buyers whose $\tau_i$ is zero are always active. Let us assume that for every buyer $i$, $\tau_i > 0$, and study the other corner cases separately in Theorem 3.7. Then, for every buyer $i$, we can define $\frac{b_i}{\tau_i}$ as a canonical threshold. Whenever $\sum_{j \in [m]} w_j p_j$ is at most this threshold, buyer $i$ is active, and she is inactive otherwise. Note that $\sum_{j \in [m]} w_j p_j$ is independent of buyers, and only depends on the price vector $\mathbf{p}$. Hence, for any price vector $\mathbf{p}$, let us define $\mathbf{W}(\mathbf{p}) = \sum_{j \in [m]} w_j p_j$. Therefore, for any price vector $\mathbf{p}$ where $\frac{b_i}{\tau_i} \geq \mathbf{W}(\mathbf{p})$, that buyer is active. As mentioned, for now we assume we have $n$ buyers with positive minimum thresholds, and they are ordered as

$$\frac{b_1}{\tau_1} \geq \frac{b_2}{\tau_2} \geq \cdots \geq \frac{b_n}{\tau_n} \geq 0.$$

Given this, for any price vector $\mathbf{p}$ such that $\frac{b_i}{\tau_i} \geq \mathbf{W}(\mathbf{p}) > \frac{b_{i+1}}{\tau_{i+1}}$, the set of active buyers is $\{1, \dots, i\}$; moreover, if $\frac{b_n}{\tau_n} \geq \mathbf{W}(\mathbf{p})$, then all buyers are active. Note that as $\mathbf{W}(\mathbf{p})$ increases continuously, the set of active buyers shrinks monotonically. Figure 1 depicts these thresholds and the corresponding sets of active buyers. For price vectors with $\mathbf{W}(\mathbf{p}) > \frac{b_1}{\tau_1}$, no buyer is active; consequently, no trade occurs and every seller's revenue is zero. While trivial equilibria exist in which all sellers set arbitrarily large prices—so that no trade occurs and no seller can profit from a unilateral deviation—such equilibria are not in the interest of any party involved. Hence, from now on we focus on price profiles under which trade occurs. We define

$$\mathcal{P} = \left\{ \mathbf{p} \in \mathbb{R}_+^m, \quad \sum_{j \in [m]} w_j p_j \leq \frac{b_1}{\tau_1} \right\}$$

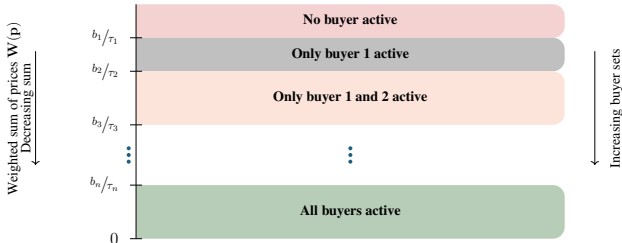

*Figure 1.* Buyer activity regions as a function of the weighted sum of prices $\mathbf{W}(\mathbf{p}) = \sum_{j \in [m]} w_j p_j$.

as the set of all price profiles for which at least one buyer is active, and therefore a trade happens. We restrict our attention to finding Nash Equilibria in $\mathcal{P}$.

**State.** We refer to any price vector $\mathbf{p}$ satisfying $\mathbf{W}(\mathbf{p}) = \frac{b_i}{\tau_i}$ as being in *State $i$* (or as *corresponding to State $i$*). Hence, we refer to $\frac{b_i}{\tau_i}$ as the threshold of state $i$; moreover, for any $i < j$ with $i, j \in [n]$, we have $\frac{b_i}{\tau_i} \geq \frac{b_j}{\tau_j}$, and we say state $i$ is *higher than or equal to* state $j$; otherwise, we say that $i$ is *lower* (or *below*) $j$. Note that as we move to lower states, the corresponding canonical threshold decreases. Consequently, the set of active buyers strictly increases (see Figure 1).

States play a central role in our analysis: as we show in the next claim, any equilibrium—if it exists—must correspond to one of these states. Consequently, it suffices to restrict attention to price vectors that lie in some state, which substantially narrows the set of candidates for equilibrium.

**Claim 3.1.** *Given any set of prices $\mathbf{p} \in \mathcal{P}$ where $\mathbf{W}(\mathbf{p}) \neq \frac{b_i}{\tau_i}$ for all $i \in [n]$, all sellers gain higher revenue by (slightly) increasing their price.*

As an immediate corollary, we obtain the following structural property: from any non-equilibrium price vector, any revenue-maximizing deviation must land on a state (i.e., make the weighted price hit some threshold $\frac{b_i}{\tau_i}$). In particular, every profitable deviation ends at one of the $n$ states.

**Corollary 3.2.** *Given any price vector $\mathbf{p}$ that is not a NE, a revenue-maximizing seller $j$ will only deviate to a price $p'_j$ such that*

$$\sum_{k \in [m] \setminus \{j\}} w_k p_k + w_j p'_j = \frac{b_i}{\tau_i} \quad \text{for some } i \in [n].$$

The following claim shows that the total seller revenue is fixed in each state, and the revenue share of any seller $j$, is proportional to $w_j p_j$.

**Claim 3.2.** *Given any price vector $\mathbf{p}$ corresponding to state $i$, the total seller revenue depends only on the state, denote it by $T(i)$. Accordingly, seller $j$'s revenue under a price*

*vector* $\mathbf{p}$ *in state* $i$ *is given by:*

$$R_j(p) = w_j p_j \left( \frac{T(i)\tau_i}{b_i} \right).$$

According to Claim 3.2, in each state $i$, the total seller revenue can be divided equally among all sellers by setting $p_j = \frac{b_i}{w_j m \tau_i}$ for any seller $j$. We will call this price vector as CONSTPROD prices in state $i$.

**Definition 3.3** (CONSTPROD prices). A vector of prices $\mathbf{p} = (p_1, \cdots, p_m)$ is a Constant Product pricing or CONSTPROD pricing if for any pair of sellers $j$ and $j'$, $p_j w_j = p_{j'} w_{j'}$. We denote a CONSTPROD price vector at state $i$ by $\mathbf{p}^c(i)$. Further, note that the price of $j^{th}$ seller's data in $\mathbf{p}^c(i)$ is $p_j^c(i) = \frac{b_i}{m \tau_i w_j}$.

CONSTPROD prices are particularly appealing because they satisfy a useful symmetry (proved in the next section): for a given state, either every seller has an incentive to deviate or no seller does. Consequently, to verify whether a CONSTPROD price vector in a state is a Nash equilibrium, it suffices to check the deviation incentive of a single seller; if that seller does not deviate, then none do, and the price vector constitutes a Nash equilibrium. To characterize deviations across states, fix a price vector $\mathbf{p}$ at state $i$. For any other state $k$, define

$$p_j' = p_j + \frac{1}{w_j} \left( \frac{b_k}{\tau_k} - \frac{b_i}{\tau_i} \right). \tag{3}$$

If $p_j' \geq 0$, then seller $j$ can deviate from state $i$ to state $k$ by changing her price to $p_j'$. Indeed, holding all other sellers' prices fixed, the resulting price vector $\mathbf{p}'$ satisfies $W(\mathbf{p}') = \frac{b_k}{\tau_k}$.

Our goal is to show that the CONSTPROD prices at some state constitute a Nash equilibrium. To do so, we introduce the *graph-of-deviations* as follows. To define it we only consider the distinct states.

**Definition 3.4** (Graph-of-deviations). Let the graph-of-deviations $G$ be a directed graph with $n'$ nodes, each representing one of the $n' \leq n$ *distinct* states. In particular, if two states have same $\frac{b_i}{\tau_i}$, we represent them as only node in the graph-of-deviations. If, under the CONSTPROD prices in state $i$, a seller finds it profitable to deviate to state $k$, we include a directed edge from node $i$ to node $k$ in $G$. We refer to this edge as a *deviation arc* from state $i$ to state $k$, denoted by $i \rightarrow k$. See Figure 2 for a brief illustrative example and Appendix B for a comprehensive example.

In graph-of-deviations, if a node has no outgoing edge, its corresponding state is called a *sink* state.

**Claim 3.3.** CONSTPROD *prices in sink states should be a Nash Equilibrium.*

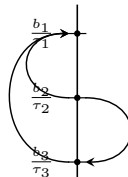

*Figure 2.* An example for graph-of-deviations with three states. A deviation arc from a state $i$ to $j$ implies that a seller would want to deviate from CONSTPROD prices at state $i$ to a new price at state $j$. Since state 1 does not have any outgoing arc, CONSTPROD prices at state 1 form a Nash Equilibrium.

*Proof.* By Claim 3.1, any profitable deviation must move the price vector to a state. Therefore, if there is a node with no outgoing edge in the graph-of-deviations, this means that CONSTPROD prices in the corresponding state has no profitable deviation to a distinct state. Further, a unilateral deviation to a state with the same $\frac{b}{\tau}$ value cannot happen since the sum cannot remain the same if only one seller changes her price. Therefore, from a sink state of the graph-of-deviations, there are no profitable deviations to any other state implying that it is a Nash equilibrium. $\square$

In the next section, we prove that the graph-of-deviations always contains a sink; consequently, a Nash equilibrium always exists.

### 3.2. Equilibrium Existence

**Lemma 3.5.** *Fix* CONSTPROD *prices at state* $i$ *and consider any other state* $k$. *There exists a seller who has a profitable deviation from state* $i$ *to state* $k$ *if and only if the state-only inequality*

$$\mathsf{Ineq}(i,k) \qquad : \qquad \frac{T(i)}{T(k)} \;<\; m + (1-m)\frac{b_i \tau_k}{b_k \tau_i}. \tag{4}$$

*holds, where* $T(\cdot)$ *is the total seller revenue in the corresponding state.*

**Implication.** Under CONSTPROD prices at state $i$, if $\mathsf{Ineq}(i,k)$ holds then *every* seller has an incentive to deviate from $i$ to $k$; otherwise, *no* seller does. A deviation arc $i \rightarrow k$ is present in the graph of deviations if and only if $\mathsf{Ineq}(i,k)$ holds.

**Theorem 3.6.** *In a data market with homogeneous buyers who have Leontief valuations and strictly positive minimum thresholds* $\tau_i > 0$ *for all buyers* $i$, *a Nash equilibrium exists.*

*Proof Sketch.* First, we show that for any distinct states $i \neq k$, the two inequalities $\mathsf{Ineq}(i,k)$ and $\mathsf{Ineq}(k,i)$ (defined in Equation (4)) cannot both hold. Consequently, the graph-of-deviations contains no 2-cycles.

Second, we show that for any three distinct states $i, j, k$ with $j$ between $i$ and $k$ (i.e., $i < j < k$ or $i > j > k$), $\mathsf{Ineq}(i,k)$

implies $\mathsf{Ineq}(i,j)$ or $\mathsf{Ineq}(j,k)$. Equivalently, if the "long" deviation arc $i \to k$ exists, then for *every* intermediate state $j$ between $i$ and $k$, there is a "short" deviation arc either $i \to j$ or $j \to k$. For example, in Figure 2, the presence of a deviation arc from state 3 to state 1 implies that there must also be a deviation arc $3 \to 2$ or $2 \to 1$.

Finally, suppose the graph contains a directed cycle of length greater than 2. Using the preceding property, we can transform it into a strictly shorter directed cycle; iterating this shortening procedure must eventually produce a 2-cycle, contradicting the fact that no 2-cycles exist. Hence, the graph-of-deviations is acyclic, and since every directed acyclic graph contains at least one sink, the graph has a sink state. By Claim 3.3, the CONSTPROD prices corresponding to sink states form Nash equilibria. □

In Theorem 3.6, we show that a Nash equilibrium exists when all buyers have strictly positive minimum thresholds. The next theorem handles the remaining corner cases—when only a (possibly empty) subset of buyers have positive thresholds—and completes the existence picture.

**Theorem 3.7.** *In a data market with homogeneous buyers who have Leontief valuations, where only a subset of buyers have strictly positive minimum thresholds, either a Nash equilibrium exists—provided the total budget of zero-threshold buyers is sufficiently small—or no finite Nash equilibrium exists; moreover, which case holds can be determined in polynomial time.*

### 3.3. Best Equilibrium

We now prove a structural lemma that applies to *any* Nash equilibrium. Let $S$ denote the set of sink states in the graph-of-deviations. We show that $S$ contains every state that can admit a Nash equilibrium under *any* pricing scheme. Since, by Claim 3.1, any Nash equilibrium must correspond to one of the $n$ states, it follows that sink states in the graph-of-deviations fully characterize all Nash equilibria.

**Lemma 3.8.** *Let $s \in [n]$ be a state that admits a Nash equilibrium under some pricing scheme. Then $s$ should be a sink in the graph-of-deviations.*

Let $s^*$ denote the sink state with the minimal corresponding threshold, i.e. $s^* \in \arg\min_{i \in \mathcal{S}} \left\{ \frac{b_i}{\tau_i} \right\}$, breaking ties arbitrarily. As we show in the next theorem, the CONSTPROD prices in state $s^*$ enjoy several desirable properties.

**Theorem 3.9.** *Among all Nash equilibria, the CONSTPROD prices at the sink state with the smallest threshold simultaneously (i) maximize total seller revenue, (ii) maximize buyer welfare, (iii) maximize buyer participation, (iv) are equitable across sellers, and (v) are computable in polynomial time.*

*Proof Sketch.* By Lemma 3.8, every Nash equilibrium must

correspond to a sink state in the graph-of-deviations. We first show that for any sink state, the total seller revenue and buyer welfare are fixed by the state itself and are independent of the particular price vector. We then compare sink states and prove that the sink state with the minimum threshold, denoted $s^*$, attains the highest total seller revenue and total buyer welfare. Consequently, every equilibrium price vector corresponding to $s^*$, including the CONSTPROD prices at $s^*$, maximizes total revenue and welfare among all equilibria. Moreover, since $s^*$ has the smallest threshold, it has the maximum number of active buyers among all equilibria. From Claim 3.2, seller revenues are equitable under CONSTPROD. Finally, constructing the graph-of-deviations and finding the sink states take $O(n^2)$, so $s^*$ and the CONSTPROD prices corresponding to it can be computed in polynomial time. □

**Implication.** A mediating platform can compute the best equilibrium by constructing the graph-of-deviations. By Lemma 3.5, this construction only requires checking whether $\mathsf{Ineq}(i,k)$ holds for each pair of states $(i,k)$. Interestingly, $\mathsf{Ineq}(i,k)$ is independent of the proportion vector $w$; thus, to compute the best equilibrium the platform only needs the buyers' budgets and their minimum thresholds.

### 3.4. Best Response Dynamics

In this section we study how the market evolves under decentralized seller behavior, i.e., without any platform intervention. For a price vector $\mathbf{p} = (p_1, \ldots, p_m)$, let $\mathbf{p}_{-j}$ denote the prices of all sellers other than $j$. Given $\mathbf{p}$, seller $j$ best responds by choosing a new price $p'_j$ that maximizes her revenue holding $\mathbf{p}_{-j}$ fixed, yielding the updated vector $(\mathbf{p}_{-j}, p'_j)$. Starting from an initial price vector $\mathbf{p}^1$, the corresponding best-response dynamics generate a (possibly infinite) sequence $\{\mathbf{p}^1, \mathbf{p}^2, \ldots\}$ where at each step an arbitrary seller updates her price to a best response to the current vector.

We prove that from any initial prices, these best-response dynamics converge to a $(1+\epsilon)$-approximate Nash equilibrium in $O\left(\frac{L}{\epsilon}\right)$ steps; throughout, we assume $L$ is polynomial in the input size. In Section 5, we empirically compare these decentralized outcomes, in terms of welfare and total revenue, to the best equilibrium on synthetic data.

**Theorem 3.10.** *Suppose a data market with homogeneous buyers who have Leontief valuations and strictly positive minimum thresholds $\tau_i > 0$ for all buyers $i$, where sellers update via $(1+\epsilon)$-improving deviations: a seller changes her price only if a unilateral deviation increases her revenue by a factor of at least $1+\epsilon$. Then the resulting best-response dynamics converge to a $(1+\epsilon)$-approximate Nash equilibrium in $O\left(\frac{L}{\epsilon}\right)$ steps.*

*Proof sketch.* Consider any sequence of $q$ deviations that

returns to the same state. We show that along such a sequence the product of seller prices increases by a factor of at least $(1 + \epsilon)^q$. This yields an $O\left(\frac{L}{\epsilon}\right)$ bound on the number of deviations between two visits to the same state. Applying the same argument across all states bounds the total number of deviations in any best-response dynamics by $O\left(\frac{L}{\epsilon}\right)$, proving convergence.

## 4. Substitutable Data

In this section, we study markets with linear utilities. While equilibria may fail to exist in general, we show that an equilibrium always exists with a single buyer which extends to platforms that allow price discrimination. See Appendix D for all missing proofs of Sections 4.1 and 4.2.

### 4.1. Non-Existence with multiple buyers

**Theorem 4.1.** *In a data market with $m$ sellers and $n$ buyers, where buyers have linear utility over the sellers' datasets, no equilibrium may exist.*

*Proof Sketch.* We construct an instance with two sellers and 101 buyers in which no equilibrium exists. The first 100 buyers have budget 1, the remaining buyer has budget 10, and all buyers have identical linear valuations for the two datasets. We show that for every possible price profile, at least one seller has a profitable deviation. Intuitively, while the prices are such that $p_1 + p_2 > 1$, the sellers first undercut each other to capture the demand of the 100 small budget buyers. As soon as the prices reach $p_1 + p_2 = 1$, the sellers have incentive to deviate to a larger price to capture the leftover budget of the large budget buyer. An exhaustive analysis of price regimes shows us that no price vector is stable. The argument extends to buyers with strictly positive minimum utility requirements by choosing these thresholds sufficiently small, ensuring all deviations remain feasible. □

A stronger version of the above theorem, which asserts that even approximate Nash equilibria do not exist in a data market with $m$ sellers and $n > 1$ buyers appeared independently in (Chaudhury et al.).

### 4.2. Existence under Price Discrimination

Given the strong non-existence result for linear utilities, we next identify conditions under which equilibrium can be restored. We show that with a single buyer, an equilibrium always exists, which naturally motivates price discrimination across buyers on a platform.

**Theorem 4.2.** *In a market with a single buyer and $m$ sellers, where the buyer has linear preferences over datasets, an equilibrium always exists. Moreover, an equilibrium that simultaneously maximizes total seller revenue, buyer welfare,*

*and is fair to sellers can be computed in polynomial time.*

*Proof Sketch.* Suppose the buyer's budget is $b$ and her value for the $j^{th}$ seller's dataset is $w_j$, her minimum utility requirement is $u^{min}$. If $u^{min} > \sum_{j \in [m]} w_j$, the buyer never participates in the market and any pricing profile is an equilibrium and by default an optimal one. Otherwise, we set prices as $p_j = \frac{w_j b}{\sum_{j \in [m]} w_j}$. At these prices, no seller can profitably deviate. This yields a stable, revenue and welfare optimal equilibrium that allocates revenue proportionally to seller values and is thereby fair to the seller.

**Corollary 4.3.** *With multiple buyers and linear utilities, an equilibrium exists when sellers can price discriminate: each seller sets prices for each buyer independently, so the problem decomposes into separate single-buyer instances. Furthermore, a revenue-optimal, welfare-optimal, and seller-fair equilibrium can be computed in polynomial time.*

### 4.3. Integral Datamarkets

Although equilibria may not exist in general integral linear data markets, they do exist with a single buyer or under price discrimination and maximize revenue and welfare. However, these equilibria can be highly unfair, in sharp contrast to our earlier markets where optimality and fairness coexist. Formal results are in Appendix E.

### 4.4. Practical Insight

> For linear preferences, platforms should implement price discrimination to achieve optimal equilibrium.

## 5. Empirical Evaluation

We provide a comprehensive empirical assessment of our model in both complementary and substitutable data markets. We perform the following experiments:

- **Leontief datamarkets with synthetic datasets (Section 5.1).** We construct synthetic datamarkets with 100 sellers and 100 buyers with thresholds $\tau_i = 0.5$ and budgets drawn uniformly from $[0.5, 1]$.
- **Leontief datamarkets with real-world datasets (Section 5.2).** We use the federated learning benchmarks given by FLamby (Ogier du Terrail et al., 2022) to simulate datasets that exhibit complementary behavior. The market parameters are determined from these datasets.
- **Linear datamarkets with synthetic datasets (Appendix F.3).** Finally we run experiments with a single linear buyer with budget 1 and $\tau = 0$; and 100 sellers. We generate each buyer's valuation for each seller independently at random from $[0.001, 1]$. The detailed results of this are in Appendix F.3.

**Evaluation Metrics.** For each setup, we evaluate three aspects. First, we check whether the best-response dynamics converge – column Dev gives the number of deviations after which the BRD converges. Second, for each market instance, different equilibria may be reached depending on the initialization. For Leontief data markets with homogeneous buyers, Theorem 3.9 allows us to compute the optimal equilibrium—namely, the equilibrium that maximizes both revenue and welfare. We therefore run BRD from the zero-price initialization and compare the welfare and revenue of the resulting equilibrium with those of this optimal equilibrium. For all other settings, we empirically estimate the best revenue and welfare attainable at equilibrium by running BRD from 100 random initializations, where each initial price is drawn uniformly from $[0, 1/m]$, and report the ratios $\frac{\text{average welfare}}{\text{best welfare}}$, and analogously for revenue $\frac{\text{average revenue}}{\text{best revenue}}$. Finally, we aggregate these statistics across multiple random market instances. Our results are as follows.

## 5.1. Leontief Datamarkets with Synthetic Data

- For homogeneous buyers we assume that the sellers are unit-weight ($w_j = 1$) and we generate 1000 random instances, aggregating the results as mentioned above. The results are in Table 1.
- To incorporate heterogeneity, in each instance and for each buyer $i$, we independently draw weights $w_\ell^i$ for seller $\ell$ uniformly at random, and then rescale them so that $\max_{\ell \in [m]} w_\ell^i = 1$. The results are in Table 2.

*Table 1.* Equilibrium quality ratios for synthetic data with homogeneous buyers (100 sellers, 100 buyers, 1000 random instances).

| Stat. | BR/max $W$ | BR/max $R$ | Dev |
|---|---|---|---|
| Min | 0.0706 | 0.0751 | 50 |
| Max | 1.0000 | 1.0000 | 1340 |
| Mean | 0.7681 | 0.7696 | 436.9 |
| Std. | 0.3172 | 0.3153 | 195.9 |

*Table 2.* Equilibrium quality ratios for synthetic data with heterogeneous buyers (100 buyers, 100 sellers, 100 instances × 100 initial prices).

| Stat. | BR/max $W$ | BR/max $R$ | Dev |
|---|---|---|---|
| Min | 0.350 | 0.105 | 3 |
| Max | 1.000 | 1.000 | 550 |
| Mean | 0.571 | 0.552 | 69.167 |
| Std. | 0.132 | 0.147 | 77.308 |

**Observations.** We observe that in all instances, the BRD always converged to an equilibrium. The quality of equilibrium, as measured by the seller revenue and buyer welfare, depends on the initial price vector we start with.

## 5.2. Leontief Datamarkets with Real-World Data

We use the **FLamby** benchmark (Ogier du Terrail et al., 2022), which provides real-world, cross-silo FL datasets with natural label skew. Among its offerings, three – **Fed-Heart Disease**, **Fed-ISIC**, and **Fed-Camelyon16** involve classification tasks and are therefore directly applicable to our market model. Full details on how to design the market using these benchmarks are in Appendix F. We note that the buyers here are heterogeneous. Our results for the Fed-Heart Disease dataset are in Table 3. The remaining results are in Appendix F.2.

*Table 3.* Equilibrium quality ratios for **Fed-Heart Disease** (4 buyers, 2 sellers, 100 instances × 100 initial prices).

| Stat. | BR/max $W$ | BR/max $R$ | Dev |
|---|---|---|---|
| Min | 0.056 | 0.090 | 0 |
| Max | 1.000 | 1.000 | 7 |
| Mean | 0.239 | 0.238 | 2.634 |
| Std. | 0.205 | 0.164 | 0.744 |

**Observation.** Across all datasets, we observe that the BRD always converges to an equilibrium. The quality of this equilibrium depends on the price vector we start with.

## 5.3. Experimental Insights for Leontief Datamarkets

In all the experiments with homogeneous or heterogeneous buyers, we found that the BRD always converges to an equilibrium. This hints towards our conjecture that for complementary datasets, there probably exist non-trivial equilibria. Further, the BRD might not converge to optimal equilibria, therefore the existence of a mediating platform driving it towards optimal equilibria is recommended for Leontief datamarkets.

# 6. Discussion

We study equilibrium-based pricing for data markets. For homogeneous buyers with perfectly complementary preferences (Leontief), we establish strong positive results, including existence, efficient computation, and welfare– and fairness–optimal equilibria. For perfectly substitutable preferences, we identify fundamental non-existence barriers and show that optimal equilibria can nevertheless be achieved under price discrimination. Together, these results provide principled guidance for platforms on how to price datasets when buyer values are known. We view this work as a foundational step. Real data markets will typically feature buyers whose preferences lie between pure complementarity and pure substitutability. Extending our framework to such hybrid settings, including CES-type utilities, is a natural next direction, and we believe our techniques offer a strong starting point for these markets.

## Impact Statement

As AI increasingly relies on data, understanding how to price and allocate datasets is critical for building efficient, fair, and sustainable marketplaces that allows access to high-quality data to everyone at reasonable price. Our work provides a rigorous analysis of equilibrium pricing in data markets, highlighting conditions under which efficiency, revenue, and fairness can be simultaneously achieved. While we focus on idealized settings – perfect complementarity (Leontief) or perfect substitutability (linear), our results still reveal fundamental trade-offs and lay a foundation for future research in more complex, real-world data markets. Additionally, we do not foresee any negative impact of this work.

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

## A. Additional Related Work.

Below we expand on the related work on Nash equilibrium, pricing games, and markets.

- **Nash Equilibrium.** Nash equilibrium (NE) (Nash et al., 1950) is a fundamental notion in game theory, ensuring stability in the presence of strategic agents. Computability of NE has been studied extensively within CS since last three decades for various games. For many natural games, the problem turned out to be hard, for example, PPAD-hard for two-player matrix games (Chen et al., 2009) even under special structures like sparsity and binary-payoff (Abbott et al., 2005; Chen et al., 2006; Mehta, 2014). In light of these negative results, polynomial-time computability of NE in our game is an important step forward.

- **Markets.** For the traditional non-rivalrous goods, various markets have been studied within both Economics and Computer Science, for the existence and computation of equilibrium. Some of the well-studied models are of Fisher (Devanur et al., 2002; Orlin, 2010) and Arrow-Debreu (exchange market) (Ye, 2005; Chen et al., 2009), where the sellers are price takers due to the large market assumption. This leads to competitive equilibrium (CE) as a solution concept, where when buyers demand their optimal bundle, the aggregate demand meets supply for each good. These markets have been studied extensively under both Leontief and Linear utility functions (for example, see (Devanur et al., 2002; Codenotti & Varadarajan, 2004; Orlin, 2010; Garg & Végh, 2019; Garg, 2017)). Notably, in the Fisher market, the allocation rule requires that the total allocation of goods should be at most one (assuming normalization), meaning the same item cannot be allocated to multiple buyers. However, in data markets, the same piece of data can be sold to multiple buyers, and therefore, the main difference between traditional and data markets is that the allocation of data is not bounded by one unit. Concisely, in a data market, there is no allocation constraint on the side of sellers.

  In the absence of the large market assumption, the relevant models are $(i)$ monopoly (Machlup, 1937), where a single seller sets prices to maximize her revenue, and $(ii)$ oligopoly (Allen & Hellwig, 1986), where a small set of buyers compete in a pricing game to maximize their revenue. Our model aligns best with the oligopolistic market.

- **Pricing Games.** Pricing games are well-studied in Economics literature under various modes of competition and pricing. Some of the popular forms are: $(i)$ Bertrand competition, where firms are selling the same product and the customer buys from the cheapest firm, leading to marginal cost as the equilibrium prices. $(ii)$ Differentiated Bertrand, where firms products are slightly different. Therefore, competition is lesser than the Bertrand model and prices can go above their marginal production costs. $(iii)$ Stackelberg Pricing, where a leader firm chooses the price and the followers then choose their price. This is a model for situations where an initial firm exists in the market and different firms are then entering the market. The firms that enter later can observe the pricing and production decisions of the leading firm and accordingly set prices.

## B. Comprehensive Example

Suppose there are two data sellers. Seller 1 provides only cat images, and seller 2 provides only dog images. There are also two buyers, both of whom face the same learning task: training a classifier to distinguish cats from dogs. Test samples are assumed to be drawn i.i.d. with uniform class priors. To avoid class imbalance, each buyer wishes to acquire equal amounts of cat and dog images. Assuming the two sellers offer datasets of equal size, this means that each buyer demands the same fraction from both sellers. Hence both buyers have the same weight vector $\mathbf{w} = (w_1, w_2)$ with $w_1 = w_2 = 1$.

Each buyer must also procure a minimum fraction of data from each seller in order to achieve statistically reliable performance. Buyer 1 has the more relaxed accuracy requirement, so she needs at least a $\frac{1}{4}$ fraction of each dataset; thus $\tau_1 = \frac{1}{4}$. Buyer 2 has the stricter accuracy requirement, so she needs at least a $\frac{1}{3}$ fraction of each dataset; thus $\tau_2 = \frac{1}{3}$. Assume both buyers have the same budget of one dollar:

$$b_1 = b_2 = 1.$$

By Definition 3.1, buyer $i$ is *active* if she can afford to purchase at least the minimum required amount $\tau_i w_j$ from every seller $j$. Therefore buyer $i$ is active if and only if

$$\tau_i \sum_{j \in \{1,2\}} w_j p_j \leq b_i.$$

Since $b_1 = b_2 = 1$ and $w_1 = w_2 = 1$, Buyer 1 is active if and only if $p_1 + p_2 \leq 4$, whereas Buyer 2 is active if and only if $p_1 + p_2 \leq 3$. Hence:

- if $p_1 + p_2 \le 3$, both buyers are active;
- if $3 < p_1 + p_2 \le 4$, only buyer 1 is active;
- if $p_1 + p_2 > 4$, no buyer is active.

Let $A(p)$ denote the number of active buyers under price vector $p = (p_1, p_2)$.

Under the utility specification, every active buyer demands the same proportion from both sellers. Thus, for a given price vector $p$, the demand of any active buyer from either seller is

$$x_{ij} = \min\left(\frac{1}{p_1 + p_2}, 1\right).$$

It follows that seller $j$'s revenue is

$$R_j(p) = A(p)\, p_j \min\left(\frac{1}{p_1 + p_2}, 1\right). \tag{5}$$

Whenever $p_1 + p_2 \notin \{3, 4\}$, the set of active buyers does not change by slightly changing the prices, so $A(p)$ is locally constant. Therefore, each seller can strictly increase revenue by raising her own price. This is formally proved by Claim 3.1. Consequently, if an equilibrium exists, it must lie on one of the threshold hyperplanes

$$p_1 + p_2 = 3 \quad \text{or} \quad p_1 + p_2 = 4.$$

We therefore define two states as follows:

- **state 1**: $p_1 + p_2 = 4$,
- **state 2**: $p_1 + p_2 = 3$.

Now apply Definition 3.3. In each state, CONSTPROD prices satisfy

$$p_j w_j = p_{j'} w_{j'} \qquad \text{for all sellers } j, j'.$$

Since $w_1 = w_2 = 1$, CONSTPROD prices simply split the total price equally across sellers. Hence:

$$p^c(1) = (2, 2) \qquad \text{for state 1,} \qquad p^c(2) = (1.5, 1.5) \qquad \text{for state 2.}$$

The graph of deviations has one node for each state, so here it has two nodes. By Definition 3.4, if under the CONSTPROD prices of state $i$ a seller can profitably deviate to a price vector in state $k$, then we draw a directed edge from node $i$ to node $k$. Because the sellers are symmetric, it suffices to examine seller 1.

First consider CONSTPROD prices in **state 1**, namely $p^c(1) = (2, 2)$. At these prices, only Buyer 1 is active, so $A(p^c(1)) = 1$. Using (5), seller 1's revenue is

$$R_1(p^c(1)) = 1 \cdot 2 \cdot \frac{1}{4} = \frac{1}{2}.$$

Suppose seller 1 deviates from state 1 to state 2. Since every price vector in state 2 satisfies $p_1 + p_2 = 3$, and seller 2's price remains fixed at 2, seller 1 must set $p_1' = 1$. Thus the deviating price vector is $p' = (1, 2)$. Under this deviation, both buyers are active, so $A(p') = 2$. Therefore seller 1's new revenue is

$$R_1(p') = 2 \cdot 1 \cdot \frac{1}{3} = \frac{2}{3},$$

which is strictly larger than $\frac{1}{2}$. Hence seller 1 has a profitable deviation from state 1 to state 2. Therefore the graph of deviations contains an edge from state 1 to state 2, so state 1 is not a sink. By Claim 3.3, CONSTPROD prices in state 1 cannot be a Nash equilibrium.

Now consider CONSTPROD prices in **state 2**, namely $p^c(2) = (1.5, 1.5)$. At these prices, both buyers are active, so $A(p^c(2)) = 2$. Seller 1's revenue is

$$R_1(p^c(2)) = 2 \cdot 1.5 \cdot \frac{1}{3} = 1.$$

Suppose seller 1 deviates from state 2 to state 1. Since every price vector in state 1 satisfies $p_1 + p_2 = 4$, and seller 2's price remains 1.5, seller 1 must set $p'_1 = 2.5$. Thus the deviating price vector is $p' = (2.5, 1.5)$. At these prices only buyer 1 remains active, so $A(p') = 1$. Hence seller 1's revenue becomes

$$R_1(p') = 1 \cdot 2.5 \cdot \frac{1}{4} = \frac{5}{8},$$

which is strictly less than her current revenue of 1. By symmetry, the same conclusion holds for seller 2. Therefore no seller wants to deviate from state 2 to state 1.

It remains to rule out deviations from state 2 to prices outside the two state boundaries. If seller 1 raises her price while keeping $p_1 + p_2 \in (3, 4]$, then only buyer 1 remains active throughout that interval, so the set of active buyers is unchanged. By Claim 3.1, revenue is then maximized at the highest such price, namely at the boundary $p_1 + p_2 = 4$, i.e. in state 1. But we have already shown that even this deviation is unprofitable. If seller 1 raises her price further so that $p_1 + p_2 > 4$, then all buyers become inactive and her revenue falls to zero. Finally, if seller 1 lowers her price below the state-2 level, then both buyers remain active, but her revenue falls because she is reducing price while the active set is unchanged.

Therefore seller 1 has no profitable deviation from CONSTPROD prices in state 2, and by symmetry neither does seller 2. Hence, CONSTPROD prices in state 2 form a Nash equilibrium. This also agrees with Claim 3.3, which states that CONSTPROD prices at sink states are Nash equilibria. Figure 3 illustrates the graph-of-deviations for this example.

State 1: $p_1 + p_2 = 4$        CONSTPROD prices of state 1: $p^c(1) = (2, 2)$

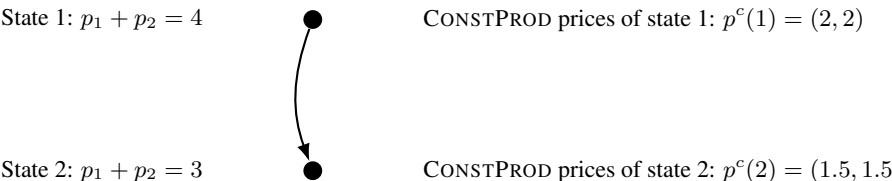

State 2: $p_1 + p_2 = 3$        CONSTPROD prices of state 2: $p^c(2) = (1.5, 1.5)$

*Figure 3.* Graph of deviations.

# C. Appendix for Section 3

## C.1. Additional Notations and Preliminaries

Before we proceed with the proofs, we need to define additional notations. In a state $i$, let us partition the active buyers into two groups, $G_1$ and $G_2$, where $G_1$ contains the buyers whose budget strictly exceeds $\frac{b_i}{\tau_i}$, and $G_2$ contains the rest of the buyers. Let us define $t_i = |G_1|$ as the total number of buyers in $G_1$ and $B'_i = \sum_{a \in G_2} b_a$ as the total budget of active buyers in $G_2$.

For each state $i$, define a function $f : [n] \to \mathbb{R}^+$ by

$$f(i) = t_i + \frac{B'_i}{b_i / \tau_i}.$$

Given a price vector $p = (p_1, \dots, p_m)$ in state $i$, we prove the following claim about the revenue of a seller $j \in [m]$.

**Claim C.1.** *The revenue of seller $j$ under a price vector $p = (p_1, \dots, p_m)$ corresponding to state $i$ is*

$$R_j(p) = w_j p_j \left( t_i + \frac{B'_i}{b_i / \tau_i} \right) = w_j p_j f(i),$$

*and the total seller revenue in state $i$ is*

$$T(i) = \frac{b_i}{\tau_i} f(i).$$

*Proof.* For any buyer $a \in G_1$, as $b_a > \frac{b_i}{\tau_i}$, and since $\frac{b_i}{\tau_i} = \sum_{j \in [m]} w_j p_j$, buyer $a$ can purchase the maximum amount needed from every seller, in otherwords $x_{aj} = w_j$ for all $j$. Therefore, every buyer in $G_1$ will pay $w_j p_j$ to the seller $j$. Therefore, the revenue of seller $j$ from buyers in $G_1$ is $w_j p_j \cdot t_i$.

For any buyer $a \in G_2$ we have that $\frac{b_a}{\tau_a} \geq \sum_{j \in [m]} w_j p_j \geq b_a$ since she is active, and she is not a member of $G_1$. Therefore, these buyers spend their full budget in the market, and their optimal decision to maximize their utility (defined in Equation (1)), is to set their demand from every seller $j$ as

$$x_{aj} = \frac{b_a w_j}{\sum w_j p_j} = \frac{b_a w_j}{b_i / \tau_i}$$

since, we have $\sum w_j p_j = \frac{b_i}{\tau_i}$. As a result, the revenue of seller $j$ from buyers in $G_2$ can be written as $w_j p_j \cdot \frac{B_i'}{b_i / \tau_i}$.

Putting it together, the revenue of seller $j$ in a price $p$ corresponding to state $i$ can be written as

$$R_j(p) = w_j p_j \left( t_i + \frac{B_i'}{b_i / \tau_i} \right) = w_j p_j f(i).$$

The total seller revenue equals $\sum_j w_j p_j f(i)$. Since $\frac{b_i}{\tau_i} = \sum_j w_j p_j$, we can rewrite the total revenue in state $i$ as

$$T(i) = \frac{b_i}{\tau_i} f(i),$$

which depends only on the state and not on the particular price vector. □

## C.2. Why Kakutani's Fixed-Point Theorem Does Not Apply

### C.2.1. NON-CONVEXITY OF BEST-RESPONSE

Consider an example with two buyers and two sellers, where both sellers assign the same weight to each buyer, i.e., $w_1 = w_2 = 1$. The buyers share identical minimum thresholds, $\tau_1 = \tau_2 = 0.5$, but have different budgets: $b_1 = 2$ and $b_2 = 1$. Using the notations and claims developed in Section 2, we demonstrate that if the second buyer sets her price to $p_2 = 1$, then the first buyer's best-response is to choose either $p_1 = 1$ or $p_1 = 3$; any other price is suboptimal. This illustrates that the best-response set is not convex.

Given this setup, there are two relevant states:

- **State 1**: $p_1 + p_2 = \frac{b_1}{\tau_1} = 4$
- **State 2**: $p_1 + p_2 = \frac{b_2}{\tau_2} = 2$

In state 1, only buyer 1 is active, and her budget is fully exhausted. Thus,

$$f(1) = \frac{b_1}{\frac{b_1}{\tau_1}} = \tau_1 = 0.5.$$

In state 2, both buyers are active, and their budgets are fully exhausted. Therefore,

$$f(2) = \frac{b_1 + b_2}{\frac{b_2}{\tau_2}} = 1.5.$$

Now, given that seller 2 has fixed her price at $p_2 = 1$, Claim 3.1 implies that, in order to maximize her revenue, seller 1 must choose a price $p_1$ such that the resulting price vector $p = (p_1, 1)$ corresponds to either state 1 or state 2. Otherwise, she can strictly increase her revenue by raising her price, rendering any other pricing strategy suboptimal.

- If $p_1 = 1$, the price vector $p = (1, 1)$ corresponds to state 2, and by Claim C.1, seller 1's revenue is:

$$r_1(p, 2) = 1 \cdot f(2) = 1 \cdot 1.5 = 1.5.$$

- If $p_1 = 3$, the price vector $p = (3, 1)$ corresponds to state 1, and seller 1's revenue is:

$$r_1(p, 1) = 3 \cdot f(1) = 3 \cdot 0.5 = 1.5.$$

Since seller 1's revenue is the same under both pricing strategies, her best-response set is $\{1, 3\}$, which is evidently non-convex.

C.2.2. BEST-RESPONSE MAY CHANGE DISCONTINUOUSLY WITH OTHERS' PRICES

Consider the same example from the previous subsection, where seller 1 selects $p_1 = 1$ as her best-response to $p_2 = 1$. Suppose now that seller 2 slightly increases her price to $p_2 = 1 + \epsilon$ for some $\epsilon > 0$. Then, by Claim 3.1, seller 1 must again choose a price $p_1$ such that the resulting price vector $p = (p_1, 1 + \epsilon)$ corresponds to either state 1 or state 2; otherwise, she can strictly increase her revenue by increasing her price.

- If $p_1 = 1 - \epsilon$, then $p = (1 - \epsilon, 1 + \epsilon)$ corresponds to state 2. By Claim C.1, seller 1's revenue is:

$$r_1(p, 2) = (1 - \epsilon) \cdot f(2) = (1 - \epsilon) \cdot 1.5 = 1.5 - 1.5\epsilon.$$

- If $p_1 = 3 - \epsilon$, then $p = (3 - \epsilon, 1 + \epsilon)$ corresponds to state 1, and seller 1's revenue is:

$$r_1(p, 1) = (3 - \epsilon) \cdot f(1) = (3 - \epsilon) \cdot 0.5 = 1.5 - 0.5\epsilon.$$

Since $r_1(p, 1) > r_1(p, 2)$ for any $\epsilon > 0$, seller 1's best-response becomes $p_1 = 3 - \epsilon$. Thus, as seller 2's price increases infinitesimally from $p_2 = 1$ to $p_2 = 1 + \epsilon$, seller 1's best-response jumps discontinuously from $p_1 = 1$ to $p_1 = 3 - \epsilon$. This demonstrates that the best-response function is discontinuous with respect to others' prices.

## C.3. Missing Proofs of Section 3.1

**Claim 3.1.** *Given any set of prices* $\mathbf{p} \in \mathcal{P}$ *where* $\mathbf{W}(\mathbf{p}) \neq \frac{b_i}{\tau_i}$ *for all* $i \in [n]$, *all sellers gain higher revenue by (slightly) increasing their price.*

*Proof.* Partition the active buyers at price vector $p = \mathbf{p}$ into two groups $G_1$ and $G_2$, where

$$G_1 = \left\{ a : b_a > \sum_{k \in [m]} w_k p_k \right\} \quad \text{and} \quad G_2 = \text{(the remaining active buyers)}.$$

Let $t = |G_1|$ be the number of active buyers whose budgets are *not* fully exhausted at prices $p$(that is every buyer $a$ where $b_a \geq \sum w_j p_j$), and let $B' = \sum_{a \in G_2} b_a$ be the total budget of the remaining active buyers in $G_2$. The seller $j$'s revenue from the first group of active buyers who fully purchase $w_j$ units from her is $w_j p_j \cdot t$, and her revenue from the second group whose optimal decision is to demand $B' w_j / \sum w_j p_j$ in total from her is $w_j p_j \cdot \frac{B'}{\sum w_j p_j}$. Therefore, seller $j'$s revenue can be written as

$$R_j(p) = w_j p_j \left( t + \frac{B'}{\sum_{k \in [m]} w_k p_k} \right).$$

As $p \in \mathcal{P}$, there exists an index $i \in [n]$ such that $\frac{b_i}{\tau_i} > \sum_{k \in [m]} w_k p_k$, define

$$Y = \min \left\{ \frac{b_i}{\tau_i} : \frac{b_i}{\tau_i} > \sum_{k \in [m]} w_k p_k \right\}.$$

If $G_1 \neq \emptyset$, further update $Y$ to $\min\{Y, \min_{a \in G_1} b_a\}$.

By construction, $Y > \sum_{k \in [m]} w_k p_k$. Now fix any seller $j$ and increase her price by

$$\Delta = \frac{Y - \sum_{k \in [m]} w_k p_k}{2 w_j}, \quad \text{i.e., set} \quad p'_j = p_j + \Delta.$$

Then

$$\sum_{k \in [m]} w_k p'_k = \sum_{k \in [m]} w_k p_k + w_j \Delta = \frac{1}{2} \left( \sum_{k \in [m]} w_k p_k + Y \right) < Y,$$

so the total weighted price remains strictly below $Y$. Consequently, no buyer in $G_1$ becomes budget-exhausted (since every $a \in G_1$ satisfies $b_a \geq Y_1 \geq Y$), and the active set does not change (since we also remain below the threshold $\frac{b_i}{\tau_i}$). Therefore both the set of active buyers and the partition $(G_1, G_2)$ remain unchanged after the deviation; hence $t$, the number of buyers

in $G_1$, and $B'$, the total budget of other active buyers in $G_2$ do not change. With demand unchanged and $p_j$ strictly larger, seller $j$'s revenue strictly increases.

$\square$

**Claim 3.2.** *Given any price vector* $\mathbf{p}$ *corresponding to state* $i$, *the total seller revenue depends only on the state, denote it by* $T(i)$. *Accordingly, seller* $j$'s *revenue under a price vector* $\mathbf{p}$ *in state* $i$ *is given by:*

$$R_j(p) = w_j p_j \left( \frac{T(i)\tau_i}{b_i} \right).$$

*Proof.* By Claim C.1, the total seller revenue $T(i)$ depends only on the state $i$. Substituting $f(i) = \frac{T(i)}{b_i/\tau_i}$ into the expression for an individual seller's revenue from Claim C.1 yields the desired form and completes the proof. $\square$

### C.4. Missing Proofs of Section 3.2

We first present an alternative formulation of Lemma 3.5, which we will use repeatedly in the proofs of this section.

**Lemma C.1.** *Given* CONSTPROD *prices in state* $i$, *deviation of one seller to state* $k$ *implies the deviation of all sellers to state* $k$. *Moreover, a deviation arc from* $i \to k$ *is present iff*

$$t_i \frac{b_i}{m\tau_i} + \frac{B'_i}{m} < \left( \frac{b_i}{m\tau_i} - \left[ \frac{b_i}{\tau_i} - \frac{b_k}{\tau_k} \right] \right) \left( t_k + \frac{B'_k}{\frac{b_k}{\tau_k}} \right).$$

*Proof.* Given a seller $j$, she deviates from CONSTPROD prices $p$ in state $i$ to state $k$ iff her revenue increases. Note that under CONSTPROD prices in state $i$, for every seller $j$ we have $w_j p_j = \frac{b_i}{m\tau_i}$. Given seller's revenue by Claim C.1, she finds it profitable to deviate iff

$$t_i \frac{b_i}{m\tau_i} + \frac{B'_i}{m} < w_j p'_j \left( t_k + \frac{B'_k}{\frac{b_k}{\tau_k}} \right).$$

where $p'_j$ is the new price. As $p$ is the set of CONSTPROD prices at state $i$, we have that $\sum_{l \in [m]} w_l p_l = \frac{b_i}{\tau_i}$, and as the new state is $k$, $\sum_{l \in [m] \setminus j} w_l p_l + w_j p'_j = \frac{b_k}{\tau_k}$.

Hence, we have that $p'_j = p_j - \frac{\left[ \frac{b_i}{\tau_i} - \frac{b_k}{\tau_k} \right]}{w_j}$. The inequality above is equivalent to

$$t_i \frac{b_i}{m\tau_i} + \frac{B'_i}{m} < \left( w_j p_j - \left[ \frac{b_i}{\tau_i} - \frac{b_k}{\tau_k} \right] \right) \left( t_k + \frac{B'_k}{\frac{b_k}{\tau_k}} \right). \tag{6}$$

As we have $w_j p_j = w_{j'} p_{j'}$ for any seller $j'$, we have that

$$t_i \frac{b_i}{m\tau_i} + \frac{B'_i}{m} < \left( w_{j'} p_{j'} - \left[ \frac{b_i}{\tau_i} - \frac{b_k}{\tau_k} \right] \right) \left( t_k + \frac{B'_k}{\frac{b_k}{\tau_k}} \right) = w_{j'} \left( p_{j'} - \frac{\left[ \frac{b_i}{\tau_i} - \frac{b_k}{\tau_k} \right]}{w_{j'}} \right) \left( t_k + \frac{B'_k}{\frac{b_k}{\tau_k}} \right).$$

Note that if seller $j'$ deviates from CONSTPROD prices in state $i$ to $k$, her new price is $p'_{j'} = p_{j'} - \frac{\left[ \frac{b_i}{\tau_i} - \frac{b_k}{\tau_k} \right]}{w_{j'}}$. As

$$t_i \frac{b_i}{m\tau_i} + \frac{B'_i}{m} < w_{j'} p'_{j'} \left( t_k + \frac{B'_k}{\frac{b_k}{\tau_k}} \right),$$

seller $j'$ obtains a higher revenue by deviating from CONSTPROD prices in state $i$ to $k$ as well. As this analysis is for any seller $j'$, deviation of one seller implies deviation of all sellers when sellers are at CONSTPROD prices in state $i$.

Recall that CONSTPROD prices $p$ in state $i$ implies for all seller $j \in [m]$, $w_j p_j = \frac{b_i}{m\tau_i}$. Inserting this in Equation (6), we have that any seller finds it profitable to deviate from CONSTPROD prices at state $i$ to state $k$ iff

$$t_i \frac{b_i}{m\tau_i} + \frac{B'_i}{m} < \left( \frac{b_i}{m\tau_i} - \left[ \frac{b_i}{\tau_i} - \frac{b_k}{\tau_k} \right] \right) \left( t_k + \frac{B'_k}{\frac{b_k}{\tau_k}} \right).$$

□

Now we prove Lemma 3.5.

**Lemma 3.5.** *Fix* CONSTPROD *prices at state $i$ and consider any other state $k$. There exists a seller who has a profitable deviation from state $i$ to state $k$ if and only if the state-only inequality*

$$\mathsf{Ineq}(i,k) \qquad : \qquad \frac{T(i)}{T(k)} \;<\; m + (1-m)\frac{b_i\tau_k}{b_k\tau_i}. \tag{4}$$

*holds, where $T(\cdot)$ is the total seller revenue in the corresponding state.*

*Proof.* From Lemma C.1, we have that a deviation arc from $i \to k$ is present iff

$$t_i\frac{b_i}{m\tau_i} + \frac{B_i'}{m} < \left(\frac{b_i}{m\tau_i} - \left[\frac{b_i}{\tau_i} - \frac{b_k}{\tau_k}\right]\right)\left(t_k + \frac{B_k'}{\frac{b_k}{\tau_k}}\right).$$

Using Claim C.1, we have $T(i) = \frac{b_i}{\tau_i}f(i)$ with $f(i) = t_i + \frac{B_i'}{b_i/\tau_i}$. Thus, the left-hand side equals $\frac{T(i)}{m}$, while the right-hand side can be rewritten as

$$\frac{T(k)}{m}\left(m + (1-m)\frac{b_i\tau_k}{b_k\tau_i}\right),$$

which establishes the claim and completes the proof. □

First, we want to prove that graph-of-deviations does not have a cycle of size 2. In other words, there is no $i,j \in [n]$ such that both $i \to j$ and $j \to i$ deviation arcs exist.

**Lemma C.2.** *Graph-of-deviations does not have a cycle of size* 2.

*Proof.* Consider any two states $i$ and $j$, where $i < j$, implying state $i$ is above state $j$. Suppose a seller deviates from CONSTPROD prices at state $i$ to state $j$. Then by Lemma C.1, and letting $\Delta = \frac{b_i}{\tau_i} - \frac{b_j}{\tau_j}$ we have that

$$\left(\frac{b_i}{m\tau_i} - \Delta\right)\left(\frac{b_jt_j + \tau_j B_j'}{b_j}\right) > t_i\frac{b_i}{m\tau_i} + \frac{B_i'}{m} = \frac{b_it_i + B_i'\tau_i}{m\tau_i}$$

Which is true iff

$$\Delta < \frac{b_i}{m\tau_i} - \frac{b_j(b_it_i + B_i'\tau_i)}{m\tau_i(b_jt_j + B_j'\tau_j)}$$

Similarly, a deviation from $j$ to $i$ occurs iff:

$$\left(\frac{b_j}{m\tau_j} + \Delta\right)\left(\frac{b_it_i + \tau_i B_i'}{b_i}\right) > \frac{b_jt_j + B_j'\tau_j}{m\tau_j}$$

which is iff:

$$\Delta > \frac{b_i(b_jt_j + B_j'\tau_j)}{m\tau_j(b_it_i + B_i'\tau_i)} - \frac{b_j}{m\tau_j} \tag{7}$$

Therefore, if there is a cyclic deviation then we necessarily need:

$$\frac{b_i(b_jt_j + B_j'\tau_j)}{m\tau_j(b_it_i + B_i'\tau_i)} - \frac{b_j}{m\tau_j} < \frac{b_i}{m\tau_i} - \frac{b_j(b_it_i + B_i'\tau_i)}{m\tau_i(b_jt_j + B_j'\tau_j)}$$

Let $b_it_i + B_i'\tau_i = \alpha$ and $b_jt_j + B'j\tau_j = \beta$. Therefore, for a cyclic deviation, the following must be true:

$$\frac{b_i\beta}{\tau_j\alpha} - \frac{b_j}{\tau_j} < \frac{b_i}{\tau_i} - \frac{b_j\alpha}{\tau_i\beta}$$

$$\iff \frac{b_i}{\tau_i}\left(1 - \frac{b_j\alpha}{b_i\beta}\right) > \frac{b_j}{\tau_j}\left(\frac{\beta b_i}{\alpha b_j} - 1\right)$$

$$\iff \frac{b_i}{\tau_i}\frac{b_i\beta - b_j\alpha}{\beta b_i} > \frac{b_j}{\tau_j}\frac{b_i\beta - b_j\alpha}{\alpha b_j} \tag{8}$$

$$\iff \frac{b_i\beta - b_j\alpha}{\beta\tau_i} > \frac{b_i\beta - b_j\alpha}{\alpha\tau_j}.$$

**Claim C.2.** *If there is a deviation arc from state $i$ to a lower state $j$ then $\alpha\tau_j < \beta\tau_i$ and $b_i\beta > b_j\alpha$.*

*Proof.* Given CONSTPROD prices $p$ at state $i$, suppose a seller $u$ deviates to a lower state $j$ by changing her price from $p_u$ to $p'_u$. Since $\sum_{l\in[m]} w_l p_l = \frac{b_i}{\tau_i}$ and $\sum_{l\in[m]\setminus\{u\}} w_l p_l + w_u p'_u = \frac{b_j}{\tau_j}$, it follows that

$$w_u(p'_u - p_u) = \frac{b_j}{\tau_j} - \frac{b_i}{\tau_i}.$$

Because state $i$ is above state $j$, we have $\frac{b_j}{\tau_j} < \frac{b_i}{\tau_i}$, implying $p'_u < p_u$. Therefore, seller $u$ reduces her price and gains higher revenue in state $j$. Using Claim C.1 to compute seller revenue, we obtain

$$w_u p'_u\left(t_j\frac{b_j}{m\tau_j} + \frac{B'_j}{m}\right) > w_u p_u\left(t_i\frac{b_i}{m\tau_i} + \frac{B'_i}{m}\right).$$

Noting that $p'_u < p_u$, this inequality implies that

$$t_j\frac{b_j}{\tau_j} + B'_j > t_i\frac{b_i}{\tau_i} + B'_i \tag{9}$$

As $\frac{\beta}{\tau_j} = t_j\frac{b_j}{\tau_j} + B'_j$ and $t_i\frac{b_i}{\tau_i} + B'_i = \frac{\alpha}{\tau_i}$, this proves $\beta\tau_i > \alpha\tau_j$.

Now considering inequality 9 with $\frac{b_i}{\tau_i} > \frac{b_j}{\tau_j}$, we can multiply the same sides of the inequalities and obtain

$$\frac{b_i}{\tau_i}\left(t_j\frac{b_j}{\tau_j} + B'_j\right) > \frac{b_j}{\tau_j}\left(t_i\frac{b_i}{\tau_i} + B'_i\right)$$

As, $\frac{b_i\beta}{\tau_i\tau_j} = \frac{b_i}{\tau_i}\left(t_j\frac{b_j}{\tau_j} + B'_j\right)$ and $\frac{b_j}{\tau_j}\left(t_i\frac{b_i}{\tau_i} + B'_i\right) = \frac{b_j\alpha}{\tau_i\tau_j}$ we obtain $b_i\beta > b_j\alpha$. $\square$

Using Claim C.2 in Equation 8, we get a contradiction. Therefore, we cannot have a cycle of deviations with size 2. $\square$

**Lemma C.3.** *For any three states $i, j, k$ where $j$ is the state between $i$ and $k$, i.e. either $i < j < k$ or $i > j > k$, the existence of a deviation arc from $i \to k$ implies the presence of a deviation arc from either $i \to j$ or $j \to k$.*

*Proof.* By Lemma C.1, a deviation from $i \to k$ implies:

$$t_i\frac{b_i}{m\tau_i} + \frac{B'_i}{m} < \left(\frac{b_i}{m\tau_i} - \left[\frac{b_i}{\tau_i} - \frac{b_k}{\tau_k}\right]\right)\left(t_k + \frac{B'_k\tau_k}{b_k}\right).$$

Recall that for any state $u \in [n]$, $f(u) = t_u + \frac{B'_u\tau_u}{b_u}$. Hence, we can write the inequality above as

$$\frac{b_i}{m\tau_i}f(i) < \left(\frac{b_i}{m\tau_i} - \left[\frac{b_i}{\tau_i} - \frac{b_k}{\tau_k}\right]\right)f(k) \iff \frac{f(i)}{f(k)} < \frac{(\frac{b_i}{m\tau_i} - [\frac{b_i}{\tau_i} - \frac{b_k}{\tau_k}])}{b_i/m\tau_i}. \tag{10}$$

Suppose for contradiction, that $i \nrightarrow j$ and $j \nrightarrow k$. Hence we must have:

$$\frac{b_i}{m\tau_i}f(i) \geq \left(\frac{b_i}{m\tau_i} - \left[\frac{b_i}{\tau_i} - \frac{b_j}{\tau_j}\right]\right)f(j) \iff \frac{f(i)}{f(j)} \geq \frac{(\frac{b_i}{m\tau_i} - [\frac{b_i}{\tau_i} - \frac{b_j}{\tau_j}])}{b_i/m\tau_i}.$$

and

$$\frac{b_j}{m\tau_j} f(j) \geq \left(\frac{b_j}{m\tau_j} - \left[\frac{b_j}{\tau_j} - \frac{b_k}{\tau_k}\right]\right) f(k) \iff \frac{f(j)}{f(k)} \geq \frac{\left(\frac{b_j}{m\tau_j} - \left[\frac{b_j}{\tau_j} - \frac{b_k}{\tau_k}\right]\right)}{b_j/m\tau_j}.$$

By multiplying the two inequalities above we have that

$$\frac{f(i)}{f(k)} \geq \frac{\left(\frac{b_i}{m\tau_i} - \left[\frac{b_i}{\tau_i} - \frac{b_j}{\tau_j}\right]\right)}{b_i/m\tau_i} \cdot \frac{\left(\frac{b_j}{m\tau_j} - \left[\frac{b_j}{\tau_j} - \frac{b_k}{\tau_k}\right]\right)}{b_j/m\tau_j}.$$

Combining this with Equation (10), we have that

$$\frac{\left(\frac{b_i}{m\tau_i} - \left[\frac{b_i}{\tau_i} - \frac{b_k}{\tau_k}\right]\right)}{b_i/m\tau_i} > \frac{\left(\frac{b_i}{m\tau_i} - \left[\frac{b_i}{\tau_i} - \frac{b_j}{\tau_j}\right]\right)}{b_i/m\tau_i} \cdot \frac{\left(\frac{b_j}{m\tau_j} - \left[\frac{b_j}{\tau_j} - \frac{b_k}{\tau_k}\right]\right)}{b_j/m\tau_j}.$$

Let $I = \frac{b_i}{\tau_i}$, $J = \frac{b_j}{\tau_j}$ and $K = \frac{b_k}{\tau_k}$. By multiplying the above inequality by $m$ we get:

$$IJ(1-m) + mKJ > IJ(1-m)^2 + m(1-m)IK + m(1-m)J^2 + m^2KJ$$

By simplifying the terms above we get this is equivalent to

$$(K-J)(I-J) > 0.$$

As we set $I = \frac{b_i}{\tau_i}$, $J = \frac{b_j}{\tau_j}$ and $K = \frac{b_k}{\tau_k}$, this implies

$$\left(\frac{b_k}{\tau_k} - \frac{b_j}{\tau_j}\right)\left(\frac{b_i}{\tau_i} - \frac{b_j}{\tau_j}\right) > 0. \tag{11}$$

However, since $j$ is the middle state and the graph-of-deviations only has distinct states, we must have either $i > j > k$, which implies $\frac{b_k}{\tau_k} > \frac{b_j}{\tau_j} > \frac{b_i}{\tau_i}$, or $i < j < k$, which implies $\frac{b_k}{\tau_k} < \frac{b_j}{\tau_j} < \frac{b_i}{\tau_i}$. In both cases, the terms $\left(\frac{b_k}{\tau_k} - \frac{b_j}{\tau_j}\right)$ and $\left(\frac{b_i}{\tau_i} - \frac{b_j}{\tau_j}\right)$ have opposite signs, which contradicts Equation (11). Therefore, the existence of a deviation arc from $i \to k$ implies the presence of a deviation arc from either $i \to j$ or $j \to k$. □

We are now ready to prove the following key lemma.

**Lemma C.4.** *Graph-of-deviations is acyclic.*

*Proof.* Suppose, for contradiction, that the graph-of-deviations contains a cycle $c$. By Lemma C.2, the cycle must have length greater than 2, i.e., $|c| > 2$. Let $H$ denote the highest state visited in this cycle, and let $B$ be the state with an edge to $H$ in the cycle, while $A$ is the state such that $H \to A$ is an edge in the cycle. Since $|c| > 2$, we must have $A \neq B$.

We now consider the following two cases:

1. **Case $A > B$:** This implies that state $A$ lies below state $B$. By Lemma C.3, the existence of a deviation arc from $H \to A$ implies the existence of a deviation arc either from $H \to B$ or from $B \to A$. The first case would imply a 2-cycle between $H$ and $B$, contradicting Lemma C.2. Therefore, the arc $H \to A$ must imply the existence of the arc $B \to A$. Replacing the edges $\{B \to H, H \to A\}$ with $\{B \to A\}$ in $c$ yields a strictly shorter cycle.
2. **Case $A < B$:** This implies that state $A$ lies above state $B$. By Lemma C.3, the existence of a deviation arc from $B \to H$ implies the existence of a deviation arc either from $B \to A$ or from $A \to H$. The second case would again imply a 2-cycle between $H$ and $A$, which is ruled out by Lemma C.2. Hence, the arc $B \to H$ must imply the existence of the arc $B \to A$, and replacing $\{B \to H, H \to A\}$ with $\{B \to A\}$ yields a shorter cycle.

In either case, we construct a strictly shorter cycle. By repeating this process, we must eventually arrive at a cycle of size 2, which contradicts Lemma C.2. Therefore, the graph-of-deviations cannot contain any cycles.

□

We now prove the following theorem.

**Theorem 3.6.** *In a data market with homogeneous buyers who have Leontief valuations and strictly positive minimum thresholds $\tau_i > 0$ for all buyers $i$, a Nash equilibrium exists.*

*Proof.* As established by Lemma C.4, the graph-of-deviations is acyclic, and therefore must contain at least one sink—that is, a node with no outgoing edges. By Claim 3.3, this node must be a Nash equilibrium proving the theorem. □

The next theorem completes the picture by fully characterizing equilibrium existence. Note that its proof relies on Lemma 3.8 and Theorem 3.9, which are proved in the next section; however, to remain consistent with the order of theorems in the main body, we present it here.

**Theorem 3.7.** *In a data market with homogeneous buyers who have Leontief valuations, where only a subset of buyers have strictly positive minimum thresholds, either a Nash equilibrium exists—provided the total budget of zero-threshold buyers is sufficiently small—or no finite Nash equilibrium exists; moreover, which case holds can be determined in polynomial time.*

*Proof.* Let

$$\tau_0 := \{\, i \in [n] : \tau_i = 0 \,\}$$

denote the set of buyers with zero minimum threshold, and let

$$B' := \sum_{a \in \tau_0} b_a$$

be their total budget. Buyers in $\tau_0$ are always active, and given their utility function, each buyer $a \in \tau_0$, sets their demand $x_{aj} = w_j \cdot \min\left(\frac{b_a}{\sum w_k p_k}, 1\right)$ from each seller $j$. Therefore, the revenue of seller $j$ coming only from buyers in $\tau_0$ is

$$R_j^0(p) = p_j w_j \cdot \sum_{a \in \tau_0} \min\left(\frac{b_a}{\sum w_k p_k}, 1\right). \tag{12}$$

Now we separately consider two cases:

- If every buyer is in $\tau_0$, then the total revenue of seller $j$ at any price vector $p$ equals $R_j^0(p)$. As observed, by increasing $p_j$, the revenue of seller $j$ increases, and therefore there is no finite equilibrium in this case.
- If some buyers have positive thresholds, let each buyer with positive minimum threshold define a state, construct the graph-of-deviations corresponding to these sellers, and let $s^*$ denote the sink state with the minimum corresponding threshold (the lowest sink state). From Theorem 3.9, the CONSTPROD prices in $s^*$ make the seller revenues equal; and by Claim C.1, the total revenue in that state is

$$T(s^*) = \frac{b_{s^*}}{\tau_{s^*}} f(s^*),$$

  therefore each seller's individual revenue is

$$R' = R'(p^c) = \frac{b_{s^*}}{m\tau_{s^*}} f(s^*),$$

  where $p^c$ denotes the CONSTPROD price vector in $s^*$. We now claim that if $R' \geq B'$, then $p^c$ is an equilibrium; otherwise, no finite equilibrium exists.
  Note that the maximum revenue any seller $j$ can guarantee herself from the buyers in $\tau_0$ is obtained by setting $p_j$ arbitrarily large: indeed, from Equation (12),

$$R_j^0(p) \to B' \qquad \text{as } p_j \to \infty$$

  (holding other prices fixed), since the denominator becomes dominated by $w_j p_j$. Since $s^*$ is a sink state, sellers do not have an incentive to deviate to other states, so their only remaining incentive for deviation is to set their prices arbitrarily large to capture almost all of the budget $B'$ from $\tau_0$. If $R' \geq B'$, then under the CONSTPROD prices in $s^*$, each seller's revenue already weakly exceeds what she can asymptotically extract from $\tau_0$, therefore they do not have an incentive to increase their prices, and $p^c$ is an equilibrium.

Otherwise, if $R' < B'$, there is always an incentive for every seller at the CONSTPROD prices in $s^*$ to increase her price arbitrarily so that her revenue gets arbitrarily close to $B'$, where the new revenue exceeds $R'$. Hence CONSTPROD prices in $s^*$ cannot be an equilibrium. Moreover, by Theorem 3.9, the CONSTPROD prices in every other sink state yield weakly smaller seller revenue; in particular, each seller's revenue under CONSTPROD in any sink state is $R'' \le R' < B'$. Therefore, CONSTPROD prices in no sink state can be an equilibrium.

Now it is easy to observe that if CONSTPROD prices in no state is an equilibrium, no price at any state can be an equilibrium. Lemma 3.8 already rules out the possibility of price vectors $p$ corresponding to non-sink states being an equilibrium. Suppose by contradiction that a price vector $p$ in a sink state $i$ is an equilibrium, where $p$ is not CONSTPROD. Then there exists a seller $j$ such that

$$w_j p_j < \frac{b_i}{m\tau_i}.$$

By Claim C.1, the revenue of this seller is

$$R_j(p) = w_j p_j f(i) < \frac{b_i}{m\tau_i} f(i).$$

However, since the CONSTPROD prices in this state are not an equilibrium, the seller's revenue under CONSTPROD in state $i$ satisfies

$$\frac{b_i}{m\tau_i} f(i) < B',$$

which implies $R_j(p) < B'$. Therefore seller $j$ can raise her price arbitrarily to obtain a revenue as close as possible to $B'$ from $\tau_0$, which exceeds $R_j(p)$, contradicting that $p$ is an equilibrium. Therefore, if $R' < B'$, no finite equilibrium exists.

$\square$

## C.5. Missing Proofs of Section 3.3

**Lemma 3.8.** *Let $s \in [n]$ be a state that admits a Nash equilibrium under some pricing scheme. Then $s$ should be a sink in the graph-of-deviations.*

*Proof.* Consider a price vector $p$ satisfying $\sum_j w_j p_j = \frac{b_i}{\tau_i}$ for some $i \in [n]$, and suppose no seller wishes to deviate. If $p$ is not a CONSTPROD vector, then there is a seller $j$ with $w_j p_j > \frac{b_i}{m\tau_i}$. Hence for any $\Delta > 0$ we have

$$\frac{\frac{b_i}{m\tau_i}}{\frac{b_i}{m\tau_i} - \Delta} > \frac{w_j p_j}{w_j p_j - \Delta}.$$

Since $j$ does not deviate to a lower state $k$,

$$w_j p_j\, f(i) \ge (w_j p_j - \Delta)\, f(k), \text{ where } \Delta = \frac{b_i}{\tau_i} - \frac{b_k}{\tau_k},$$

so $\frac{w_j p_j}{w_j p_j - \Delta} \ge \frac{f(k)}{f(i)}$. Combining these:

$$\frac{\frac{b_i}{m\tau_i}}{\frac{b_i}{m\tau_i} - \Delta} > \frac{f(k)}{f(i)}$$

or equivalently,

$$f(i)\frac{b_i}{m\tau_i} \ge f(k)\left(\frac{b_i}{m\tau_i} - \Delta\right).$$

Here the left side is the revenue of a seller at constant prices at state $i$ and right side is the revenue of the seller when deviating to a lower state. Therefore, no seller has an incentive to deviate downwards.

Similarly, there exists some seller $j$ that has $w_j p_j < \frac{b_i}{m\tau_i}$. Then for any $\Delta > 0$

$$\frac{\frac{b_i}{m\tau_i}}{\frac{b_i}{m\tau_i} + \Delta} > \frac{w_j p_j}{w_j p_j + \Delta}.$$

Since $j$ does not deviate upward to state $k$,

$$w_j p_j \, f(i) \; \geq \; (w_j p_j + \Delta) \, f(k), \quad \Delta = \frac{b_k}{\tau_k} - \frac{b_i}{\tau_i},$$

so $\frac{w_j p_j}{w_j p_j + \Delta} \geq \frac{f(k)}{f(i)}$. Then following similar reasoning as above, under CONSTPROD prices in state $i$, no upward deviation is profitable.

Therefore, the CONSTPROD price vector in state $i$ constitutes a Nash equilibrium. Hence state $i$ should be a sink state (with no outgoing edge) in graph-of-deviations. $\square$

**Theorem 3.9.** *Among all Nash equilibria, the* CONSTPROD *prices at the sink state with the smallest threshold simultaneously (i) maximize total seller revenue, (ii) maximize buyer welfare, (iii) maximize buyer participation, (iv) are equitable across sellers, and (v) are computable in polynomial time.*

*Proof.* Let $s^*$ denote the sink state with the smallest threshold (lowest sink state), and let $p^c$ denote the CONSTPROD prices in that state.

- Equitable seller revenues is derived directly from the definition of CONSTPROD prices and Claim C.1.
- Maximize total seller revenue: By Claim C.1, the total revenue in a state $i$ is

$$T(i) = \frac{b_i}{\tau_i} f(i).$$

Suppose by contradiction that there exists a Nash equilibrium $p'$ whose total seller revenue exceeds $T(s^*)$. By Lemma 3.8, sink states characterize all Nash equilibria, therefore $p'$ must correspond to some sink state $s'$. Since $s^*$ is the lowest sink state, $s'$ must have a larger threshold parameter, i.e.

$$\frac{b_{s'}}{\tau_{s'}} > \frac{b_{s^*}}{\tau_{s^*}}.$$

Under the CONSTPROD prices $p^c$ in $s^*$, seller revenues are equal, hence each seller $j$ obtains

$$R_j(p^c) = \frac{T(s^*)}{m}.$$

Now consider a deviation by seller $j$ from $s^*$ to $s'$. By Equation (3), to reach $s'$ she must increase her price to

$$p'_j \; = \; p_j \; + \; \frac{1}{w_j}\left(\frac{b_{s'}}{\tau_{s'}} - \frac{b_{s^*}}{\tau_{s^*}}\right).$$

Let $p' = (p'_j, p^c_{-j})$. By Claim 3.2, seller $j$'s revenue under $p'$ is

$$R_j(p') = \left(\frac{(1-m)b_{s^*}}{m\tau_{s^*}} + \frac{b_{s'}}{\tau_{s'}}\right)\frac{T(s')}{b_{s'}/\tau_{s'}} = T(s') \cdot \left(1 - \frac{m-1}{m}\frac{b_{s^*}/\tau_{s^*}}{b_{s'}/\tau_{s'}}\right).$$

Since $\frac{b_{s'}}{\tau_{s'}} > \frac{b_{s^*}}{\tau_{s^*}}$, the multiplicative term in parentheses is strictly larger than $1/m$, and hence

$$R_j(p') > \frac{T(s')}{m}.$$

Moreover, by our contradiction assumption $T(s') > T(s^*)$, we obtain

$$R_j(p') > \frac{T(s')}{m} > \frac{T(s^*)}{m} = R_j(p^c).$$

Thus seller $j$ would profitably deviate from the CONSTPROD prices in state $s^*$ to reach state $s'$, contradicting that $s^*$ is a sink state. Therefore, the total revenue in the lowest sink state is maximal.

- Maximize buyer participation: By Lemma 3.8, every Nash equilibrium corresponds to a sink state, and $s^*$ is the sink state with the smallest threshold. Since the set of active buyers increases monotonically as we move to lower states (see Figure 1), $s^*$ has the maximum number of active buyers among all sink states. Hence, any equilibrium at $s^*$ achieves maximum buyer participation among all equilibria.
- Maximize total welfare: Consider an equilibrium price vector $p'$, which must correspond to a sink state $s'$ by Lemma 3.8. Since $s^*$ has the lowest threshold among all sink states, we have

$$\frac{b_{s^*}}{\tau_{s^*}} \leq \frac{b_{s'}}{\tau_{s'}}.$$

Therefore, every buyer who is active in $s'$ is also active in $s^*$ (see Figure 1). For any price vector $p$, an active buyer $a$ sets her demand for each seller $j$ as

$$x_{aj} = w_j \cdot \min\left(\frac{b_a}{\mathbf{W}(p)}, 1\right).$$

Moreover, as $p^c$ and $p'$ correspond to states $s^*$ and $s'$ respectively, $\mathbf{W}(p^c) = \frac{b_{s^*}}{\tau_{s^*}}$ and $\mathbf{W}(p') = \frac{b_{s'}}{\tau_{s'}}$. Since $\frac{b_{s^*}}{\tau_{s^*}} \leq \frac{b_{s'}}{\tau_{s'}}$, it follows that $\mathbf{W}(p^c) \leq \mathbf{W}(p')$, and hence for every buyer $a$ active in $s'$ and every seller $j$,

$$x_{aj}(p^c) \geq x_{aj}(p').$$

Consequently, by Equation (1), the utility of every buyer active in $s'$ is weakly higher under $p^c$. Since all such buyers are also active in $s^*$, it follows that total welfare is maximized at any equilibrium corresponding to state $s^*$.

- The graph-of-deviations has at most $n$ nodes, each corresponding to a state. To determine, for any ordered pair of nodes $(i, k)$, whether there is a deviation arc $i \to k$, we check whether $\mathsf{Ineq}(i, k)$ holds; by Lemma 3.5 this takes $O(1)$ per pair, and thus $O(n^2)$ time over all pairs. Once this DAG is constructed, every node with no outgoing edge is a sink, so all sinks can be identified in $O(n)$ time. Among the sink states, the CONSTPROD prices of the sink with the smallest threshold (the lowest sink) form the best equilibrium. Hence, the total process runs in $O(n^2)$ time.

$\square$

## C.6. Missing Proofs of Section 3.4

In a data market where for every buyer $i$, $\tau_i > 0$, we want to prove that if sellers deviate only if their revenue improves by a factor of $1 + \epsilon$, a $(1 + \epsilon)$-approximate Nash equilibrium is obtained in $O(\frac{L}{\epsilon})$ steps. Before proving these, we first mention a technicality. To obtain a linear time convergence of the best-response dynamic, we need the initial price vector to be non-zero and each individual component to be bounded from below. We show how to reach such a price vector in the following claim.

**Claim C.3.** *Starting with zero initial prices, in $2m$ iterations, we either reach a Nash Equilibrium or we reach a price vector where every seller has non-zero prices. Moreover, if we are at a non-zero price vector, the logarithm of each individual price is polynomial in input size $L$.*

*Proof.* We consider the following execution of best-response dynamics. As long as there exists a seller whose price is zero, we allow such a seller to best respond if she has a profitable deviation; otherwise, we allow a seller with a non-zero price to best respond.

We make the following observations.

(1) If a seller has price 0, then any profitable deviation must increase her price. Indeed, setting a strictly positive price yields strictly positive revenue, while decreasing the price is impossible. Moreover, such a deviation necessarily moves the system to a higher state, since state thresholds are increasing functions of prices.

(2) If a seller with non-zero price is the one deviating while some seller still has price 0, then no zero-price seller has a profitable deviation. In this case, the current state must already be the highest reachable state: otherwise, a zero-price seller could deviate upward. Consequently, any deviation by a non-zero-price seller must be downward, moving the system to a lower state.

(3) Once a seller sets a strictly positive price, she will never reduce it back to zero. Doing so would strictly reduce her revenue from a positive value to zero, and hence cannot be a best response.

Combining the above, observe that each seller can switch from price 0 to a positive price at most once. Between two such switches, at most one downward deviation by a non-zero-price seller can occur. Therefore, in at most $2m$ iterations, either all sellers have strictly positive prices, or no seller has a profitable deviation, in which case we have reached a Nash equilibrium. This proves the first part of the claim.

To see the second part, let us first define $\Delta_{i,j} = \left| \frac{b_i}{\tau_i} - \frac{b_j}{\tau_j} \right|$. Then, as the price vectors evolve, the sellers move their price between two states (by Claim 3.1). Therefore, starting from 0, the price of seller $k$ evolves as summations of $\Delta_{i,j}$ for some $i, j \in [m]$, divided by her weight, $w_k$. Further, since there are only polynomially many iterations, the logarithm of this summation will be polynomial in the input size. This proves the second part of the claim. $\qquad \square$

From now on we assume that the sellers are starting with a non-zero initial price vector.

**Theorem 3.10.** *Suppose a data market with homogeneous buyers who have Leontief valuations and strictly positive minimum thresholds $\tau_i > 0$ for all buyers $i$, where sellers update via $(1 + \epsilon)$-improving deviations: a seller changes her price only if a unilateral deviation increases her revenue by a factor of at least $1 + \epsilon$. Then the resulting best-response dynamics converge to a $(1 + \epsilon)$-approximate Nash equilibrium in $O\left(\frac{L}{\epsilon}\right)$ steps.*

*Proof.* Recall that Corollary 3.2 implies that from the first step of the best-response dynamics, sellers always update their prices to satisfy the constraint

$$\sum_j w_j p_j = \frac{b_i}{\tau_i}$$

for some $i \in [n]$; otherwise, they receive suboptimal revenue. Further, from Claim C.3, all prices of the initial price vector are non-zero (or we have already reached an exact NE). Let $p_{\min} > 0$ be the lowest price from the initial price vector.

Consider a fixed state $i$ and let the sequence of states visited via best-response before returning to state $i$ be denoted by $a_1 \to a_2 \to \cdots \to a_q$, where repetition is allowed. Let $p^{a_k}$ denote the vector of prices in state $a_k$. From each state $a_k$ to $a_{k+1}$, one seller deviates. For a seller $j$ deviating between $a_k$ and $a_{k+1}$, the revenue inequality implies:

$$(1 + \varepsilon)\, w_j p_j^{a_k} f(a_k) < w_j p_j^{a_{k+1}} f(a_{k+1}).$$

Define the gain factor

$$E_k := \frac{p_j^{a_{k+1}} f(a_{k+1})}{p_j^{a_k} f(a_k)},$$

so $E_k > 1 + \varepsilon$. Then we have

$$\prod_{k=1}^{q-1} E_k > (1 + \varepsilon)^q.$$

Let $j_1, \ldots, j_q$ be the indices of the deviating sellers in this sequence. Expanding the left-hand side of above inequality, we obtain

$$\frac{p_{j_1}^{a_2} f(a_2)}{p_{j_1}^{a_1} f(a_1)} \cdot \frac{p_{j_2}^{a_3} f(a_3)}{p_{j_2}^{a_2} f(a_2)} \cdots \frac{p_{j_q}^{a_{q+1}} f(a_{q+1})}{p_{j_q}^{a_q} f(a_q)} > (1 + \varepsilon)^q.$$

Since $a_{q+1} = a_1$ (we have returned to the same state), $f(a_{q+1}) = f(a_1)$, and the inequality simplifies to:

$$\prod_{k=1}^{q} \frac{p_{j_k}^{a_{k+1}}}{p_{j_k}^{a_k}} > (1 + \varepsilon)^q.$$

Now, group the deviations by the deviating seller. For seller $t$, define the set of her deviation steps as $\{k : j_k = t\}$ and let $T(t)$ be the number of such deviations. Let $n(t, i)$ denote the index in the sequence corresponding to the $i$-th deviation of seller $t$. Since seller $t$'s price only changes when she deviates, we can write:

$$\prod_{t=1}^{m} \left( \frac{p_t^{n(t,T(t))}}{p_t^{a_1}} \right) > (1 + \varepsilon)^q.$$

Denote $p_t^{\text{new}} = p_t^{n(t,T(t))}$ (the new price after $q$ deviations), and $p_t^{\text{old}} = p_t^{a_1}$ (the original price). Then:

$$\prod_{t=1}^m \frac{p_t^{\text{new}}}{p_t^{\text{old}}} > (1+\varepsilon)^q, \quad \text{and thus} \quad \prod_{t=1}^m \frac{w_t p_t^{\text{new}}}{w_t p_t^{\text{old}}} > (1+\varepsilon)^q.$$

Since the original and new prices after $q$ deviations correspond to the same state $i$, we have that $\sum_{t \in [m]} w_t p_t^{\text{new}} = \sum_{t \in [m]} w_t p_t^{\text{old}} = \frac{b_i}{\tau_i}$. By AM–GM inequality,

$$\prod_{t=1}^m w_t p_t \leq \left( \frac{b_i}{m \tau_i} \right)^m,$$

Given that the initial product of the terms $w_t p_t$ is at least $p_{\min}^m \prod_{t=1}^m w_t$, and that after any sequence of $q$ deviations returning to state $i$ this product increases by a factor of at least $(1+\varepsilon)^q$, while never exceeding the upper bound $\left( \frac{b_i}{m \tau_i} \right)^m$, it follows that starting from state $i$, the total number of deviations that can occur before returning to it is at most $Q^i$, that must satisfy

$$p_{\min}^m \prod_{t=1}^m w_t \cdot (1+\varepsilon)^{Q^i} \leq \left( \frac{b_i}{m \tau_i} \right)^m.$$

Taking logarithms and rearranging yields:

$$Q^i \leq \frac{m \ln\left( \frac{b_i}{m \tau_i p_{\min}} \right) - \sum_{t=1}^m \ln w_t}{\ln(1+\varepsilon)} = O\left( \frac{1}{\varepsilon} \left( m \ln\left( \frac{b_i}{m \tau_i p_{\min}} \right) - \sum_{t=1}^m \ln w_t \right) \right).$$

Therefore, the total number of deviations before convergence is bounded by:

$$\sum_{i=1}^n Q^i = O\left( \frac{1}{\varepsilon} \left( m \sum_{i=1}^n \ln\left( \frac{b_i}{\tau_i} \right) - mn \ln(m p_{\min}) - n \sum_{t=1}^m \ln w_t \right) \right).$$

Suppose, for contradiction, that the dynamic continues for $Q > \sum_{i=1}^n Q^i$ steps. Let the state after $Q$ deviations be $i$. Then this state must not have been visited before the $Q - Q^i$th deviation. Similarly, the state visited before $Q - Q^i$th deviation, denoted as $i' \neq i$, must not have been visited before the $Q - Q^i - Q^{i'}$th deviation. Continuing this argument, after $Q - \sum_{i=1}^n Q^i$ steps, we would not have visited any state, which is a contradiction.

Finally, note that the number of bounds depend on $\ln(m p_{min})$. For an arbitrary initial vector, this could be very large. However, from Claim C.3, $\ln(p_{min})$ is a polynomial in input size $L$, and we can say that our algorithm converges in $O\left( \frac{L}{\epsilon} \right)$ iterations. $\qquad\square$

## D. Appendix for Section 4

**Theorem 4.1.** *In a data market with $m$ sellers and $n$ buyers, where buyers have linear utility over the sellers' datasets, no equilibrium may exist.*

*Proof.* Consider the instance with two sellers and 101 buyers. The first 100 buyers $b_1, \ldots, b_{100}$ have a budget of \$1 and buyer $b_{101}$ has a budget of \$10. All buyers value both datasets at same value, say $v$. We first state the example with $u_i^{min} = 0$ for all buyers and then extend it to $u_i^{min}$ non-zero showing that having non-zero $u_i^{min}$s does not help in having an equilibrium in the linear case. We consider the following different regimes for the prices and show that an equilibrium cannot exist anywhere. Throughout we assume that if the prices are equal the buyers tie-break in favor of first seller and if they are unequal, without loss of generality, $p_1 < p_2$.

**Case 1: $p_1 \geq 1$ and $p_2 \geq 1$.** Here the first 100 buyers can only afford one dataset and therefore they will spend all their budget on seller 1's dataset. As a result, seller 1 gets a revenue of 100 from the first 100 buyers. Even if the other seller gets

all of $10 - p_1$ from the $101^{th}$ buyer, she has the ability to increase her revenue by slightly undercutting the first seller and grabbing the budget of the first 100 buyers. Setting her price to be $p_1 - \epsilon$ for a very small $\epsilon$ gives her a revenue close to 100 as long as $p_1 \geq 1$. Therefore, again, the sellers will keep undercutting each other and there is no equilibrium here.

**Case 2:** $^{110}/_{202} \leq p_1 \leq 1$ **and** $p_2 \geq 1$**.** Now, all buyers will first buy seller 1s dataset at a cost of $p_1$. Therefore, she gets a revenue of $101p_1$. The remaining budget is spent by the buyers on seller 2s dataset and therefore she gets a revenue of $100(1 - p_1) + 10 - p_1 = 110 - 101p_1 \leq 101p_1$ in the range of $p_1$ we assume. Therefore, the second seller will again have incentive to lower her price to a value of $p_1 - \epsilon$ for some very small $\epsilon$ and get close to the revenue that seller 1 is currently obtaining.

**Case 3:** $0 \leq p_1 \leq ^{110}/_{202}$ **and** $p_2 \geq 1$**.** In this case, we follow the same calculations as Case 2 and observe that now seller 1s revenue is less than seller 2s revenue, therefore she has incentive to increase her price to $p_2 - \epsilon$ or even to some value between $^{110}/_{202}$ and 1 to receive a larger revenue. Therefore, there is no equilibrium in this regime.

**Case 4:** $p_1 \leq 1$ **and** $p_2 \leq 1$ **and** $p_1 + p_2 > 1$**.** All sellers will first buy seller 1's dataset and then spend remaining budget on seller 2's dataset. As a result, seller 1's revenue is $101p_1$ and seller 2's revenue is $100(1 - p_1) + p_2$ – since both prices are at most 1, the $101^{th}$ buyer buys both datasets completely. Now, if $p_1 > \frac{100+p_2}{201}$, the revenue of seller 1 is higher than that of seller 2 and therefore, seller 2 has an incentive to lower her price to $p_1 - \epsilon$ for very small $\epsilon$ so that we are still in the given case. On the other hand, if $p_1 < \frac{100+p_2}{201}$, then seller 1 has an incentive to increase her price to $p_2 - \epsilon$ and obtain a revenue closer to the seller 2s revenue. Finally, if $p_1 = \frac{100+p_2}{201}$, the revenue of both sellers is equal. Additionally, the first 100 buyers are buying full dataset of seller 1 then spending remaining on seller 2. At this point, the seller 2 can increase her price to $10 - p_1$ – she maintains her revenue from the first 100 buyers and increases from the $101^{th}$ buyer. Therefore, there is no equilibrium in this case.

**Case 5:** $p_1 \leq 1$ **and** $p_2 \leq 1$ **and** $p_1 + p_2 \leq 1$**.** In this case, all buyers are able to buy all datasets completely. Therefore, the revenue of seller 1 is $101p_1$ and that of seller 2 is $101p_2$. At this stage, increasing the price by either seller to however large only increases their revenue, since the amount of money spent by the buyers on the other seller is bounded. In particular seller 1 can deviate to $10 - p_2$, maintain her revenue from the first 100 buyers and increase the one from the $101^{th}$ buyer. Therefore, there is no equilibrium in this case either.

Looking at all the cases, we can see that there is no equilibrium when both prices are greater than 1, one of them is more than 1 and other is less than 1 and when both prices are less than 1. Therefore, there is no equilibrium that will exist here.

**Extension to non-zero** $u_i^{min}s$**.** Having a non-zero $u_i^{min}$ implies that for some pricing strategies, some buyers will not be able to participate thereby potentially reducing the revenue. To circumvent this, in the above example, we can keep a very small $u_i^{min}$ for all buyers so that the deviations between the strategies mentioned above can be carried out without affecting the buyers' ability to participate in the market. In particular, we can have $u_i^{min} = v/10$ where $v$ is the value the buyers have for sellers' datasets. For any pricing strategy such that $p_1 \leq 10$ and $p_2 \leq 10$, all buyers participate consistently to the above case with $u_i^{min} = 0$. Therefore, the non-existence of Nash equilibrium within those profiles continues to hold. If one or both of the prices is above \$10 and other is at most \$10, the seller with lower price has incentive to move her price to \$10 at which point the higher priced seller's data is not sold at all. Therefore, she will lower her price to be at most \$10 and there is no equilibrium in this case. Finally, if both prices are above \$10, then the first 100 buyers cannot participate and both of the sellers have incentive to deviate to $p_j = 10$ to increase their revenue. Therefore, there is no equilibrium existence in this case either. In this way, the non-existence example extends to non-zero $u_i^{min}$. $\qquad \square$

**Theorem 4.2.** *In a market with a single buyer and $m$ sellers, where the buyer has linear preferences over datasets, an equilibrium always exists. Moreover, an equilibrium that simultaneously maximizes total seller revenue, buyer welfare, and is fair to sellers can be computed in polynomial time.*

*Proof.* Consider a single buyer with budget $b$, minimum utility requirement $u^{min}$. Let there be $m$ sellers whose datasets are valued by the buyer at $w_1, \ldots, w_m$. Then, an equilibrium will always exist. Consider the following cases.

**Case 1:** $u^{min} > \sum_{j \in [m]} w_j$**.** In this case, no matter what prices the sellers set, the buyer never participates in the market and as a result, all pricing profiles are equilibrium.

**Case 2:** $\sum_{j \in [m]} w_j \geq u^{min}$**.** Consider the pricing profile given by $p_j = \frac{w_j b}{\sum_{j \in [m]} w_j}$. At these prices, the buyer gets a utility of $\sum_{j \in [m]} w_j \geq u^{min}$ and therefore will be active. For any seller, there is no incentive to reduce her price since she is already being bought completely at a higher price, her revenue does not increase by a reduction in price. On the other hand

if any seller tries to increase her price, we note that the buyer's optimal bundle will be bought so that the different datasets are chosen in order of decreasing bang-per-buck. At the given pricing, all sellers have same bang-per-buck. Any seller who increases her price will be at the lowest bang-per-buck and therefore be the last one to be considered. Consequently, if seller $j$ deviates to a higher price, then the buyer can only spend $p_j$ on this seller since the other sellers use up $b - p_j$ of her budget. Therefore the revenue of the seller does not change and this price is an equilibrium. This proves the existence of an equilibrium.

**Properties of the Equilibrium.** We show that the aforementioned equilibria are optimal for the respective cases. In the first case, when $\sum_{j \in [m]} w_j < u^{min}$, the buyer will never buy anything from the market. So by default all equilibria have $0$ revenue and $0$ welfare and all are optimal. In the second case, when $\sum_{j \in [m]} w_j \geq u^{min}$, we show that the given equilibrium is optimal. The equilibrium is clearly revenue optimal since the the market extracts full budget $b$ of the single buyer present. Similarly, the equilibrium is welfare optimal since the buyer extracts complete welfare $\sum_{j \in [m]} w_j$ from the market. The welfare is fair to the sellers in the sense that they get a portion of the budget proportional to their value – since the sellers are substitutable, this would be a fair outcome. $\square$

**Corollary 4.3.** *With multiple buyers and linear utilities, an equilibrium exists when sellers can price discriminate: each seller sets prices for each buyer independently, so the problem decomposes into separate single-buyer instances. Furthermore, a revenue-optimal, welfare-optimal, and seller-fair equilibrium can be computed in polynomial time.*

*Proof.* For each buyer, we use the pricing from Theorem 4.2. Thereby, the total revenue extracted is maximized – the market gets all budget of all buyers who can be active, the total welfare is maximized – any buyer who can be active will receive the full welfare. Additionally, all the revenue extracted will be distributed in proportion of the $w_j$s and therefore, is fair to the sellers. $\square$

# E. Integral Data Markets

In this section, we show that all our results will extend to integral markets where data should be sold as a whole unit.

## E.1. Complementary Datasets

All of our results to *integral* markets, where each dataset can be sold only in whole units. With Leontief valuations (perfect complements), a buyer either purchases the full bundle, one unit from every seller, or purchases nothing. This integral market can be viewed as a special case of our fractional model by setting the proportion vector to all ones, $w = \mathbf{1}$, and the minimum threshold to $\tau_i = 1$ for every buyer $i$, meaning that a buyer requires one full unit from each seller to obtain positive utility (see Equation (1)). Hence the integral market is a special case of the fractional market in which all buyers have strictly positive minimum thresholds, and therefore Theorem 3.6, Theorem 3.9, and Theorem 3.10 carry over unchanged.

## E.2. Substitutable Datasets

### E.2.1. NON-EXISTENCE WITH MULTIPLE BUYERS

**Theorem E.1.** *Given a datamarket with $m$ sellers and $n$ buyers, if the buyers have linear utility over the sellers' datasets and can only buy the datasets integrally, then an equilibrium may not exist.*

*Proof.* Consider the instance where there two sellers, each with value $1$ and two buyers with budget $b_1 = 0.6$ and $b_2 = 1$. The buyers are buying either the full dataset or nothing. Therefore, since the buyers value both datasets equally, they will always first try to purchase the cheaper one. We will break ties in favor of the first seller. We now show that no equilibrium exists in this case.

**Case 1:** $p_1 > 1$ **and** $p_2 > 1$**.** In this case, no buyer has budget to purchase either of the datasets and therefore both sellers have incentive to move to a price of $\$1$ to increase their revenue. Therefore, no profile in this region can be an equilibrium.

**Case 2:** $p_1 \leq 1$ **and** $p_2 > 1$**.** In this case, seller 2's dataset is not bought by either buyer and irrespective of $p_1$, she has an incentive to deviate to a price $p_1 - \epsilon$ for some small $\epsilon$ to definitely increase her revenue. Therefore, no price profile here is an equilibrium.

We give the proof by showing that there is no equilibrium when $p_1 \neq p_2$ and that there is no equilibrium when $p_1 = p_2$.

**Case 3:** $p_1 \neq p_2$**.** By symmetry, it suffices to rule out $p_1 < p_2$. Suppose $0 < p_1 < p_2 \leq 1$ and assume, for contradiction, that $(p_1, p_2)$ is a Nash equilibrium. We further separate this into four subcases.

**Case 3a:** $p_1 + p_2 < 0.6$**.** In this case, both buyers are buying both sellers datasets completely. This does not change when either of the sellers increases their price slightly to get $p_1 + p_2 = 0.6$ and the seller's revenue increases. Therefore, there is no equilibrium here.

**Case 3b:** $p_1 + p_2 = 0.6$**.** In this case, we still have both buyers' datasets being completely purchased. Since we have $p_1 < p_2$, seller 1 has incentive to increase her price to $\min\{0.6, p_2\}$ since she will still be bought by both buyers and her revenue strictly increases. Therefore, there is no equilibrium here.

**Case 3c:** $0.6 < p_1 + p_2 \leq 1$**.** In this case, the first buyer cannot buy both datasets and will only purchase the cheaper one. Since $p_1 < p_2$, this buyer will always have incentive to increase her price to be $\min\{0.6, p_2 - \epsilon\}$ for any $\epsilon$. If $p_2 \leq 0.6$, then there will be no equilibrium in this region with $p_1 \neq p_2$ – as the region for $p_1$ is not closed. On the other hand, if $p_2 > 0.6$, $p_1$ can only be $0.6$. In this case, the second buyer can also only purchase from seller 1 and no one is buying seller 2's dataset. Therefore, seller 2 will have an incentive to lower her price, implying there is no equilibrium in this region.

**Case 3d:** $p_1 + p_2 > 1$**.** In this case, no one is purchasing seller 2's dataset (since we assume that $p_1 < p_2$) and therefore, seller 2 has an incentive to deviate to a price below seller 1's price to gain positive revenue. Therefore, there is no equilibrium in this region either.

Putting all the above cases, together, there is no equilibrium for any prices $p_1 \neq p_2$. Therefore, we consider the only remaining case below.

**Case 4:** $p_1 = p_2$**.** In this case, we assume tie-breaking that the buyers will break ties in favor of seller 1.

**Case 4a: If** $p_1 = p_2 < 0.3$**.** Then both sellers' datasets are purchased by both buyers. Either of them can increase their price so that $p_1 + p_2 = 0.6$. They will still be purchased by both buyers and strictly increase in revenue. So there is no equilibrium here.

**Case 4b:** $p_1 = p_2 = 0.3$**.** In this case, revenue of each seller is $0.6$ (they earn $0.3$ from each buyer). Now, either of them can increase their price to $0.7$. Buyer 2 will purchase them at this cost and their revenue increases even though buyer 1 no longer purchases their dataset. Therefore, this is not an equilibrium.

**Case 4c:** $0.3 < p_1 = p_2 < 0.5$ In this case, tie is broken in favor of seller 1. Therefore, she earns a revenue of $2p$ and seller 2 earns a revenue of $p$. Seller 2 therefore has incentive to deviate to $1 - p_1 > 0.5$ and increase her revenue. Therefore, there is not equilibrium here.

**Case 4d:** $p_1 = p_2 = 0.5$**.** Here seller 1's dataset is being purchased by both buyers but seller 2s dataset is being purchased only by buyer 2. While seller 2 cannot increase her price, she can make her price $0.5 - \epsilon$ for some small $\epsilon$ to ensure she is purchased by both buyers and increase her revenue. Therefore, there is no equilibrium here.

**Case 4e:** $p_1 = p_2 > 0.5$**.** In this case, neither of the buyers will purchase dataset from both the sellers. Therefore, since we tie-break in favor of seller 1 always, seller 2 is never purchased. Seller 2 will therefore have an incentive to lower her price and gain non-zero revenue. Therefore there is no equilibrium in this region.

Putting all the above cases together, we have no equilibrium in this example at either equal or unequal prices. This proves our claim.

**Extension to non-zero** $u_i^{min}$**.** Integral markets extend to non-zero $u_i^{min}$s in a more straightforward manner. We can keep $u_i^{min}$ for both buyers to be any quantity smaller than $1$. Therefore, as long as either of the sellers' dataset is being purchased, the buyers will remain active. The buyers only become inactive when no seller's dataset is affordable and this does not affect the strategic price changes of the sellers.

$\square$

### E.2.2. EXISTENCE UNDER PRICE DISCRIMINATION

We now show that with a single buyer, equilibrium will always exist and that an optimal equilibrium can be computed in polynomial time. Similar to the fractional market case, it extends to optimal equilibrium under price discrimination.

**Theorem E.2.** *In a market with $m$ sellers and $1$ buyer, when the buyer has linear valuation over the sellers' datasets and*

*will buy integrally i.e., either buy a dataset or not, an equilibrium always exists and can be computed in polynomial time. Further, this equilibrium will simultaneously maximize welfare of the buyer and total revenue of the sellers.*

*Proof.* Let the budget of buyer be $b$ and her minimum utility requirement be $u^{min}$. Let the value of the buyer for seller $j$'s dataset be $w_j$. We again consider two cases.

**Case 1:** $u^{min} > \sum_{j \in [m]} w_j$**.** In this case, irrespective of the prices set by the sellers, the buyer never receives her minimum required utility. Therefore, all prices are equilibrium prices and the revenue and welfare are both always $0$. Therefore, by default all pricing profiles are optimal equilibria.

**Case 2:** $u^{min} \leq \sum_{j \in [m]} w_j$**.** Assume without loss of generality that $w_1 \geq w_2 \geq \cdots \geq w_m$. Consider the pricing given by $p_1 = b$ and $p_j = 0$ for all $j \neq 1$. Then no seller has any incentive to deviate – seller 1 clearly earns the full budget and will not deviate. For any other seller, if they change their price unilaterally, they will not be bought since the buyer has to choose between buying this dataset vs buying seller 1's dataset and she will always choose seller 1's dataset as that is valued more. Therefore, this is an equilibrium. Further, the total revenue is maximized and the total welfare is also maximized at this pricing. □

**Corollary E.3.** *Given a market with $m$ sellers and $n$ buyers, there is an equilibrium with price discrimination that simultaneously maximizes the total welfare of the buyers and total revenue of the sellers.*

*Proof.* We simply use the appropriate pricing for appropriate buyer from Theorem E.2. This will extract full budget from the market and give full welfare to each buyer. It is therefore optimal. □

*Remark* E.4. We note that, unlike markets with fractional buying and selling or Leontief utilities, the resulting equilibrium pricing in this setting is not fair to all the sellers, despite being total-welfare and revenue maximizing. Identifying an equilibrium that simultaneously achieves optimality and fairness across sellers remains an interesting open problem. This also highlights the importance of fairness considerations in data markets, as equilibria that exist without such guarantees can be unsatisfactory from a market-design perspective.

## F. Appendix for Section 5

### F.1. Leontief Datamarkets with Synthetic Data

#### F.1.1. HOMOGENEOUS BUYERS

We conducted an empirical study on synthetic markets with $m = n = 100$, unit-weight sellers ($w_j = 1$) and buyers with thresholds $\tau_i = 0.5$ and budgets drawn uniformly from $[0.5, 1]$, generating 1000 random instances.

For each instance, we measured the ratio of revenue/welfare under best-response dynamics (from zero prices) to the maximum, capturing typical decentralized performance. Table 1 reports the min, max, mean, and standard deviation of these metrics, along with the number of deviations until convergence. As shown in Figure 4, we plot the empirical distribution of the number of seller deviations by best-response dynamics to reach a Nash equilibrium across 1000 random market instances, plotted using 20 bins. The histogram is tightly concentrated around its mean of approximately 437 deviations, indicating that convergence typically occurs within a few hundred single-seller updates, and that extreme long-runs are rare.

Figure 5 displays two overlaid histograms of revenue ratios across 1000 random instances, each plotted using 12 bins. The dark-blue bars show the ratio of the minimum equilibrium revenue to the maximum equilibrium revenue, thus quantifying the worst-case inefficiency over all equilibria. The light-blue bars show the ratio of the revenue at the equilibrium reached by best-response dynamics (from zero prices) to the maximum equilibrium revenue, capturing the typical performance of decentralized price updates. We observe that while worst-case equilibria can be significantly inefficient, best-response dynamics almost 60% of the times, reaches to the optimum revenue.

Analogously, Figure 6 presents the corresponding histograms for welfare ratios, each plotted using 12 bins. The dark-blue bars plot the minimum-to-maximum welfare ratio across all equilibria, and the light-blue bars plot the welfare achieved by best-response dynamics (from zero prices) relative to the maximum equilibrium welfare. The results mirror those for revenue, showing that even though some equilibria have bad welfare guarantees, the best-response dynamic process attains optimal welfare in almost 60% of the instances.

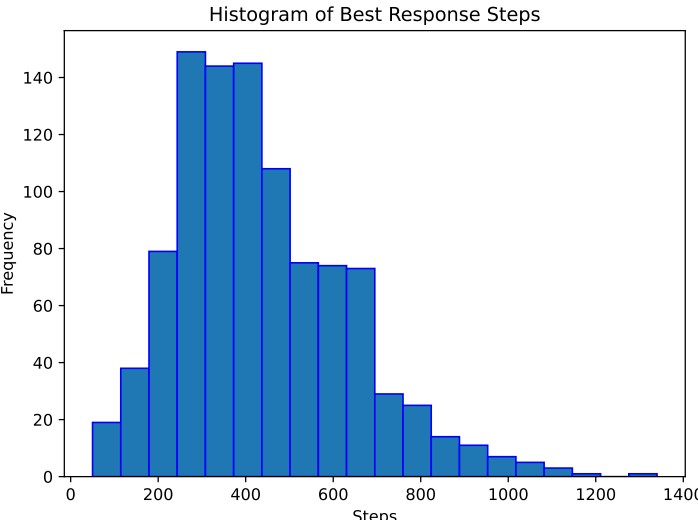

*Figure 4.* Empirical distribution of the number of single-seller deviations required by best-response dynamics to reach a Nash equilibrium, over 1000 random market instances.

**Implication.** As the statistics show, revenue and welfare under best-response dynamics can fall far short of the optimal values guaranteed by CONSTPROD prices in state $s^*$ (Theorem 3.9). These gaps—interpretable as a form of *price of anarchy*—highlight that decentralized dynamics may produce inefficient equilibria, underscoring the value of a central platform in guiding prices.

### F.1.2. HETEROGENEOUS BUYERS

We perform an empirical study on synthetic markets with $m = n = 100$, thresholds $\tau_i = 0.5$, and budgets drawn uniformly from $[0.5, 1]$, generating 100 random instances. To incorporate heterogeneity, in each instance and for each buyer $i$, we independently draw weights $w_\ell^i$ for seller $\ell$ uniformly at random, and then rescale them so that $\max_{\ell \in [m]} w_\ell^i = 1$. For each instance, we run best-response dynamics from 100 independently sampled initial price vectors, where each initial price is drawn uniformly from $[0, 1/m]$. We then evaluate the outcomes using the evaluation metrics defined in Section F.2.3. The results are reported in table 2. Similar to the earlier section, we find that best-response dynamics always converges to an equilibrium, further strengthening the conjecture that our theoretical results may extend to settings with heterogeneous buyers. Moreover, the welfare and revenue at the resulting equilibrium vary with the initial prices, further highlighting the potential value of a mediating platform that can identify an optimal equilibrium.

### F.2. Leontief Datamarkets with Real-World Data

Our goal is to instantiate the heterogeneous-buyer setting from naturally occurring label-skewed multi-center datasets and then test whether the qualitative conclusions of our theory continue to hold in this broader setting.

We use classification datasets from **FLamby** (Ogier du Terrail et al., 2022), where data are naturally partitioned across several centers and the label distributions differ across agents. This makes them particularly well suited for modeling heterogeneous buyers. Each dataset consists of $n$ centers. Each center $i \in [n]$ has its own training data $(X_{\text{train}}^i, Y_{\text{train}}^i)$ and test data $(X_{\text{test}}^i, Y_{\text{test}}^i)$, where $X$ denotes the features and $Y$ denotes the labels. Let $m$ be the total number of unique labels.

### F.2.1. DATASETS

We use the **FLamby** benchmark (Ogier du Terrail et al., 2022), which provides real-world, cross-silo FL datasets with natural label skew. Among its offerings, three involve classification tasks and are therefore directly applicable to our market model.

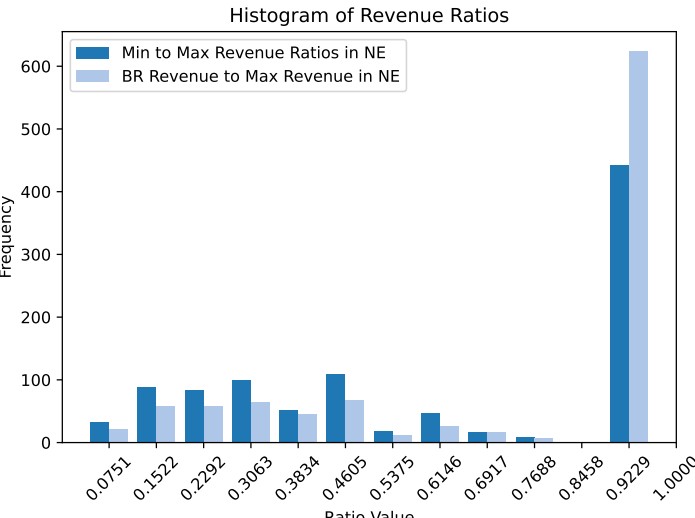

*Figure 5.* Revenue ratio distributions. Dark bars: minimum-to-maximum equilibrium revenue; light bars: revenue at the best-response equilibrium relative to the maximum equilibrium revenue.

**Fed-Heart Disease.** This dataset comprises patient records from four medical centres, where the binary label indicates the presence ($y=1$) or absence ($y=0$) of heart disease, yielding $m=2$ sellers.

**Fed-ISIC 2019.** Dermoscopic images are drawn from six clinical centres. Images are classified into $m=8$ skin-condition categories. Geographic and demographic variation across centres—differing UV-exposure levels, referral patterns, and population skin types—produces substantial label skews.

**Fed-Camelyon16.** Histopathology images from two centres are represented as extracted feature vectors, where each datapoint is labelled as *normal* or *tumour* ($m=2$), yielding two buyers and two sellers.

F.2.2. MARKET CONSTRUCTION FROM FL BENCHMARKS

In each of the datasets, the centers correspond to different medical institutions and therefore differ in terms of geography and specialty; as a result, the label skews vary significantly across centers. We study these datasets in hindsight: namely, if each center wished to purchase training datapoints for its test task, and each seller offered training datapoints from a unique label, we ask whether best-response dynamics converges, and, under different initial prices, how far apart the revenues and welfare are in the best and worst equilibria to which best-response dynamics empirically converges.

**Buyers and sellers.** We interpret each center as a *buyer* that wishes to acquire training data for its own downstream prediction task. On the supply side, we assume that there is one seller per label, and seller $\ell$ provides all available training examples of label $\ell$. Let $\mathcal{L} = \{1, \ldots, m\}$ denote the set of labels, and let

$$n_\ell = \sum_{i=1}^{n} \sum_{j=1}^{|Y_{\text{train}}^i|} \mathbf{1}\big\{(Y_{\text{train}}^i)_j = \ell\big\}$$

be the total number of training datapoints with label $\ell$ aggregated across all agents. Thus, seller $\ell$ controls a supply of size $n_\ell$.

**Buyer weights.** For each buyer $i$, we derive its preferences from the empirical label distribution of its *test* set. The motivation is that a buyer ideally wants its acquired training data to reflect the label composition of its test environment, thereby reducing distribution mismatch and the resulting training bias. Concretely, let $\mathbf{w}^i = (w_1^i, \ldots, w_m^i)$ denote the label proportions in buyer $i$'s test set, normalised so that $\max_{j \in [m]} w_j^i = 1$, where $m$ is the number of classes. More precisely, for

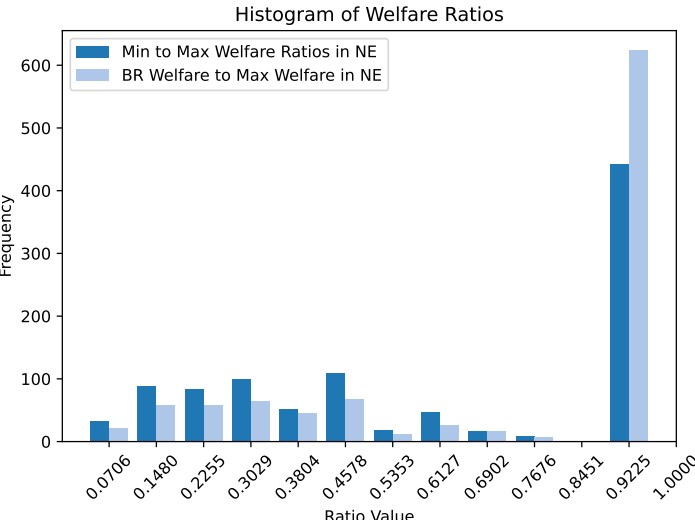

*Figure 6.* Welfare ratio distributions. Dark bars: minimum-to-maximum equilibrium welfare; light bars: welfare at the best-response equilibrium relative to the maximum equilibrium welfare.

each label $\ell \in [m]$, let

$$t_\ell^i = \sum_{j=1}^{|Y_{\text{test}}^i|} \mathbf{1}\{(Y_{\text{test}}^i)_j = \ell\}$$

be the number of test datapoints of label $\ell$ for buyer $i$. Then we define

$$w_\ell^i = \frac{t_\ell^i}{\max_{r \in [m]} t_r^i}.$$

Thus, $w_\ell^i$ captures the relative prevalence of label $\ell$ in buyer $i$'s test set, with the most frequent label assigned weight 1.

**Buyer thresholds.**    Each buyer $i$ requires at least as many training samples as the corresponding FL agent uses. Let

$$m_i = \left|Y_{\text{train}}^i\right|$$

denote that number, i.e., the total number of training samples used by FL agent $i$. Because buyer $i$ draws from seller $l$ in proportion $w_l^i$, the minimum purchase quantity from seller $l$ is $\tau_i w_l^i n_l$, where the threshold $\tau_i$ is set so that the total purchase equals $m_i$:

$$\tau_i = \frac{m_i}{\displaystyle\sum_{l=1}^{m} w_l^i n_l} . \tag{13}$$

Thus, the FL benchmark fully determines the market parameters $(\mathbf{w}^i, \tau_i)$ for every buyer $i$.

F.2.3. EXPERIMENTAL PROTOCOL

For all three datasets, the market interpretation is the same: each seller corresponds to one label and supplies all training datapoints of that label, while each center is a buyer whose required composition across labels is determined by its empirical test distribution. For each dataset, we generate 100 market instances by drawing buyer budgets independently and uniformly at random from the interval $[0.5, 1]$. For every such instance, we run best-response dynamics from 100 independently sampled initial price vectors, with each initial price drawn uniformly from $[0, 1/m]$.

**Evaluations.** We evaluate three aspects. First, we check whether the best-response dynamics converge. Empirically, in all of these experiments, every run converges to an equilibrium. This provides evidence that the convergence behavior observed in our main setting may extend to richer markets with heterogeneous buyers.

Second, for each market instance, multiple equilibria may be reached depending on initialization. We therefore quantify equilibrium variability by computing, across the equilibria obtained from the 100 initializations, the ratios

$$\frac{\text{average welfare}}{\text{best welfare}},$$

and analogously for revenue:

$$\frac{\text{average revenue}}{\text{best revenue}}.$$

Finally, we aggregate these statistics across the 100 random market instances for each dataset. The resulting tables (tables 3, 4, and 5) show that, even in these markets with heterogeneous buyers derived from real federated learning data, best-response dynamics consistently converge. At the same time, the welfare and revenue achieved at the resulting equilibria can differ substantially depending on the initial prices. These findings motivate follow-up research on establishing the existence of equilibrium in markets with heterogeneous buyers, and on studying whether platform mediation can be used to efficiently compute an optimal equilibrium in polynomial time, analogous to our current positive results under homogeneity. In the tables below, the final column ("Dev.") reports the minimum, maximum, and average number of best-response deviations required for convergence to equilibrium, aggregated over all 100 instances and 100 random price initializations per instance.

*Table 4.* Equilibrium quality ratios for **Fed-ISIC 2019** (6 buyers, 8 sellers, 100 instances $\times$ 100 initial prices).

| Stat. | BR/max $W$ | BR/max $R$ | Dev |
|-------|------------|------------|-------|
| Min   | 0.068      | 0.017      | 1     |
| Max   | 1.000      | 1.000      | 20    |
| Mean  | 0.294      | 0.469      | 4.493 |
| Std.  | 0.187      | 0.178      | 2.296 |

*Table 5.* Equilibrium quality ratios for **Fed-Camelyon16** (2 buyers, 2 sellers, 100 instances $\times$ 100 initial prices).

| Stat. | BR/max $W$ | BR/max $R$ | Dev |
|-------|------------|------------|-------|
| Min   | 0.226      | 0.499      | 0     |
| Max   | 0.989      | 1.000      | 7     |
| Mean  | 0.676      | 0.825      | 2.101 |
| Std.  | 0.266      | 0.166      | 0.923 |

## F.3. Linear Markets

We also evaluated best-response dynamics in the setting of synthetic *substitutable* data, with a single buyer and $m = 100$ sellers. The buyer has budget 1 and minimum required utility 0. We generated 100 random instances, where in each instance the buyer's values for the sellers are drawn independently and uniformly at random from $[0.001, 1]$. By Theorem 4.2 of our paper, an equilibrium price always exists in this setting, and the optimal equilibrium—which simultaneously maximizes both revenue and welfare—can be computed in polynomial time for each instance. This allows us to compute the equilibrium with highest welfare and revenue, and compare it to the equilibrium obtained by best-response dynamics when all initial prices are set to 0.001.

However, in contrast to our experiments for complementary data, where best-response dynamics always converged after at most 550 steps (deviations), we find that in the substitutable setting the dynamics often fails to converge: even after 10,000 deviations, it still does not reach an equilibrium in 19% of instances. For the instances in which it does converge to an equilibrium, we compute the ratio of the revenue and welfare achieved by best-response dynamics to the corresponding optimum, thereby capturing the typical performance of decentralized dynamics. Table 6 reports the minimum, maximum, mean, and standard deviation of these ratios, together with the number of deviations until convergence. Note that in all computed equilibria, total seller revenue exactly matches the buyer's budget, as expected. This is because whenever the total price is below the buyer's budget, sellers have an incentive to increase their prices so as to absorb the remaining budget.

All the experiments were run on a MacBook Pro with an Apple M1 Pro CPU and 16 GB RAM. The implementation uses Python 3.11, and the total time of executing this simulation was 30 minutes.

*Table 6.* Summary statistics for synthetic experiments with substitutable datasets (100 sellers, 1 buyer, 100 instances).

| Stat. | BR/Max W | BR/Max R | Devs |
|---|---|---|---|
| Min | 0.4547 | 1.000 | 491 |
| Max | 0.7221 | 1.000 | 1937 |
| Mean | 0.6021 | 1.000 | 1312 |
| Std. | 0.0573 | 0.000 | 286.9 |

