# OpenReview forum: "How to Price Data: A Market Equilibrium Based Approach"
_ICML.cc/2026/Conference — ICML 2026 regular_

### Official Review · Reviewer_Z3ab · 2026-03-08

**Soundness:** 3
**Presentation:** 3
**Significance:** 3
**Originality:** 3
**Overall Recommendation:** 5
**Confidence:** 3

**Summary:**

This paper studies equilibrium-based pricing in data markets. Recognizing that data, as a replicable and non-rivalrous good, cannot be easily priced using the classical competitive equilibrium framework designed for traditional goods, the authors propose a new pricing equilibrium model tailored for emerging data markets. The article is innovative to a certain extent, with a rigorous theoretical framework and a clear model. However, it contains too much theoretical proof and too few examples, making it difficult to read. Furthermore, the experimental section is insufficient.

**Compliance With Llm Reviewing Policy:**

Affirmed.

**Final Justification:**

The author's reply was very clear and solved all my problems. Based on the original manuscript and the author's reply, the above is our final score for this article.

**Key Questions For Authors:**

1、The experimental study is limited. The paper only reports synthetic-data simulations for best-response dynamics in Section 3.4.1, while the other core theoretical contributions lack experimental validation. Have the authors considered adding more experiments to validate the other main results and the practical performance of the proposed algorithms? For example: runtime and performance on real data, how the theoretical results deviate under heterogeneous buyers, intuitive demonstrations of equilibrium existence and non-existence under linear preferences, and equilibrium restoration under price discrimination.
2、The core of the paper is mathematical derivation and theorem proving. While this is rigorous, the proofs are numerous and dense, and there is a lack of intuitive examples, making the paper difficult to understand. Have the authors considered adding more simple numerical examples for illustration, such as how states in the graph-of-deviations are partitioned, how ConstProd prices are computed, and how an arc arises from a particular state?
3、The related work section is relatively limited and does not sufficiently cover the broader literature. In particular, the discussion of research on multiple market types, computation of Nash equilibria, pricing games, and existence proofs is not comprehensive enough. I also suggest adding more recent references to better cover the latest developments in the area.

**Limitations:**

yes

**Strengths And Weaknesses:**

Strengths
1.	The overall theoretical framework of the paper is rigorous, and the modeling is clear. In particular, the structural analysis under Leontief preferences is relatively complete, with clearly defined boundaries.
2.	The main results are logically well connected: the paper develops a coherent chain from structural results, to equilibrium existence, to best equilibrium and approximate convergence of dynamics.
3.	Terminology is used relatively consistently, and the definitions and proposition numbering are clear.
4.	The paper addresses the problem of how to reasonably price replicable and non-rivalrous data, which is of practical relevance.
5.	The results under the Leontief setting are relatively complete, and the paper also provides results within scope for the linear setting; both have theoretical and reference value.
6.	From a theoretical perspective, the paper identifies an important gap: classical equilibrium concepts do not directly apply to data markets, and it redesigns and analyzes an equilibrium model tailored to this setting. This is innovative.
7.	The paper provides a relatively complete set of theoretical results and has some significance from a mechanism design perspective.


Weaknesses
1.	The paper relies almost entirely on theoretical proofs. However, there are too many proofs and too few examples, making the paper difficult to understand and raising the barrier to entry.
2.	The experimental section is clearly limited.
3.	The exposition is overly proof-oriented: most conclusions are introduced directly in the form of theorems/lemmas, with insufficient intuitive explanations and transitions in between.
4.	The related work section is relatively limited, and does not sufficiently cover research on Nash equilibrium, pricing games, and related topics.
5.	At present, the contribution is more theoretical and conceptual than directly actionable. The limited experiments are conducted only on synthetic data, and there is no validation on real data platforms.

---

> ### Author Rebuttal · Authors · 2026-03-31
>
> We thank the reviewer for their insightful suggestions and for appreciating our model, techniques, and novelty.
>
> **Q1**:Following this suggestion, we substantially broadened the empirical evaluation to additional real-world and synthetic settings covering both complementary and substitutable data; full details are provided at https://rebuttal.tiiny.site.
>
> 1- For complementary data with heterogeneous buyers, using real healthcare datasets from FLamby as well as synthetic markets with 100 buyers and sellers, best response dynamics (BRD) always converges quickly (within O(#sellers) many steps) to a Nash equilibrium, though the equilibrium reached can vary substantially across initial prices in both seller revenue and buyer welfare, highlighting the value of a mediating platform that selects the revenue- or welfare-maximizing equilibrium.
>
> 2-  For substitutable data, in synthetic markets with one buyer and 100 sellers, behavior is qualitatively different: while complementary instances converge in seconds, similarly sized linear instances fail to converge in over 19% of cases even with 100× more time. When BRD does converge, total seller revenue equals the buyer’s budget and buyer welfare is always at least 0.45 of the optimal equilibrium welfare, suggesting that the main advantage of a mediating platform in these settings is efficient computation of the optimal equilibrium, whereas the BRD can be slow or fail to converge.
>
> Overall, these additional experiments considerably strengthen our main result: we give empirical evidence of the existence of equilibrium and convergence of BRD in markets with heterogeneous buyers under Leontief valuations.
>
> Regarding intuitive demonstrations for our results on linear preferences, we kindly note that the paper already provides a proof sketch of non-existence for linear utilities with multiple buyers immediately after Theorem 4.1, and a proof sketch of existence for linear utilities with one buyer immediately after Theorem 4.2; full proofs are given in Appendix C. For equilibrium restoration under price discrimination, the main intuition is that the platform can charge each buyer separately. Because data is replicable, allocating data to one buyer does not restrict allocation to others, so the market decomposes into independent single-buyer problems. Existence therefore follows from Theorem 4.2.
>
> **Q2**: We thank the reviewer for this helpful suggestion. We agree that simple numerical examples can significantly improve the intuition behind the graph-of-deviations framework. In response, we have added a detailed example in Section 3 of https://rebuttal.tiiny.site, and in the revision we will use it as a running example throughout Section 3 to illustrate all the main points.
> Concretely, the example considers two sellers and two buyers. Seller 1 provides only cat images and seller 2 provides only dog images, while both buyers face the same downstream task of training a cat-vs-dog classifier. Since both buyers want balanced data, they demand equal fractions from both sellers, i.e., $w_1=w_2=1$. Suppose both buyers have budget 1, but buyer 1 needs only a ¼ fraction of each dataset to be active, while buyer 2 needs a ⅓  fraction. Then buyer 1 is active when $p_1+p_2 \le 4$, while buyer 2 is active when $p_1+p_2 \le 3$. This partitions the market into two states: one with total price $4$, where only buyer 1 is active, and one with total price $3$, where both buyers are active.Under ConstProd pricing, the total price in each state is split equally across sellers, yielding candidate prices $(2,2)$ for the first state and $(1.5,1.5)$ for the second. The example then shows how the graph of deviations is formed: from $(2,2)$, a seller has a profitable deviation that moves the market to the second state, creating a directed edge from state 1 to state 2; in contrast, from $(1.5,1.5)$ no seller has a profitable deviation, so that is an equilibrium. We believe this example makes the construction and role of the graph of deviations much more transparent, and we will integrate it into the main text accordingly.
>
> **Q3**: We agree that the related work section should more comprehensively cover the broader literature. In response, we have prepared a substantially expanded related work discussion in Section 4 of https://rebuttal.tiiny.site, and we will incorporate this material into the appendix of the paper to improve comprehensiveness.
>
> **On practical insights**: While the paper is theoretical, it already yields concrete guidance: for Leontief preferences, platforms should mediate pricing to steer sellers toward fair and efficient equilibria (Section 3.5), while for linear preferences, platforms should use price discrimination to restore and compute an optimal equilibrium (Section 4.4). We will make these implications more explicit in the conclusion. More broadly, we believe that developing a strong theoretical foundation that leads to clear practical insights is highly relevant for ICML.

---

> > ### Author Rebuttal · Reviewer_Z3ab · 2026-04-02
> >
> > Thank you so much, the author clarified all my questions.

---

> > > ### Author Response · Authors · 2026-04-02
> > >
> > > Thank you very much for taking the time to consider our rebuttal. We’re glad that our clarifications and additional experiments helped address your questions.

---

### Official Review · Reviewer_bsm5 · 2026-03-11

**Soundness:** 4
**Presentation:** 3
**Significance:** 4
**Originality:** 4
**Overall Recommendation:** 5
**Confidence:** 3

**Summary:**

This paper introduces a new notion of equilibrium for data pricing based on Nash equilibrium and study it in settings where data may be complementary (Leontief) or substitutable (linear).
The authors show that equilibrium prices doesn’t exist for linear utilities even with homogeneous buyers and two sellers setting, by contrast, they establish strong existence, efficiency, and polynomial-time computation guarantees for Leontief utilities via a novel graph-of-deviations method

**Compliance With Llm Reviewing Policy:**

Affirmed.

**Key Questions For Authors:**

1. How sensitive the Leontief results under homogeneous assumption? i.e., how the results would be affected if the buyers are non-homogeneous (different w)
2. Could you give more experimental result?
3. Could you give a more precise proof of the existence of equilibrium under price discrimination for linear utility?

**Limitations:**

yes

**Strengths And Weaknesses:**

Strength
Clear motivation about why data pricing matter and why classical supply-demand equilibrium fails for data due to its non-rivalrous nature;
Introduces a Nash-equilibrium-based notion for data pricing;
Define CONSTPROD prices that satisfy symmetry that for a given state, either every seller has an incentive to deviate or no seller does;
Develops a novel proof technique graph-of-deviations that applies to any Nash equilibrium

Weakness
Assume homogeneous buyers: all buyers have the same valuation function with different budgets, and many results rely on this assumption;
Only one experiment setting on synthetic markets with m (sellers) = n (buyers) = 100, unit-weight sellers (wj = 1) and buyers with thresholds tau = 0.5;
It claim equilibrium existence under price discrimination for linear utility, but only a proof sketch without experiment

---

> ### Author Rebuttal · Authors · 2026-03-31
>
> We appreciate the reviewer’s recognition of our problem motivation, and the novelty of our proof techniques. Below we will address the questions raised by the reviewer.
>
> **Q1**:
> Our extensive additional experiments on both real-world and synthetic data show that the results should hold for the case of heterogeneous buyers as well. In particular, our experiments show that the best response dynamics (BRD) always converge quickly to an equilibrium, strongly suggesting the existence of an equilibrium and convergence of BRD in markets with heterogeneous buyers; the experimental details are provided at https://rebuttal.tiiny.site. We will add these experiments and the discussion in the final version of the paper.
>
> That said, we view homogeneity as a natural first benchmark in several data-market settings where buyers have the same downstream tasks, and even under this assumption our model already goes beyond much of the prior literature, which often considers only a single buyer [Bhawalkar et al. 2025] or even a single buyer–seller interaction [Bergemann et al. 2018, Babaioff et al. 2012]. In contrast, we study a full strategic market with multiple strategic sellers, which introduces substantial technical challenges and requires new proof techniques.
>
> **Q2**:We thank the reviewer for this comment. Following this suggestion, we substantially broadened the empirical evaluation to additional real-world and synthetic settings covering both complementary and substitutable data; full details are provided at https://rebuttal.tiiny.site.
>
> 1- For complementary data with heterogeneous buyers, we consider a market in which buyers have downstream classification tasks and purchase labeled training data from sellers, where each seller holds data for a particular label and buyers require labels in proportions determined by their test dataset. To instantiate this setting, we use three real-world healthcare datasets from FLamby to construct market parameters, and also study synthetic markets with 100 buyers and 100 sellers. Across all such instances, BRD always converges quickly (within O(#sellers) many steps) to a Nash equilibrium, though the equilibrium reached can vary substantially across initial prices in both seller revenue and buyer welfare, highlighting the value of a mediating platform that selects the revenue- or welfare-maximizing equilibrium.
>
> 2- For substitutable data, in synthetic markets with one buyer and 100 sellers, behavior is qualitatively different: while complementary instances converge in seconds, similarly sized linear instances fail to converge in over 19% of cases even with 100× more time. When BRD does converge, total seller revenue equals the buyer’s budget and buyer welfare is always at least 0.45 of the optimal equilibrium welfare, suggesting that the main advantage of a mediating platform in these settings is efficient computation of the optimal equilibrium, whereas the BRD can be slow or fail to converge.
>
> Overall, these additional experiments considerably strengthen our main result: we give empirical evidence of the existence of equilibrium and convergence of BRD in markets with heterogeneous buyers under Leontief valuations.
>
> **Q3**: Under price discrimination, seller $j$ may charge a buyer-specific price $p_{ij}$ to buyer $i$. Since data is replicable, selling to buyer $i$ does not restrict the seller from simultaneously selling the same dataset to other buyers. Therefore, the seller’s pricing decision for each buyer is independent, and the problem decomposes into separate single-buyer instances. We will add this clarification to the proof.
>
> **References**:
>
> [Bhawalkar et al. 2025]: Bhawalkar, K., Dean, J., Liaw, C., Mehta, A., and Patel, N. Equilibria and Learning in Modular Marketplaces. EC 2025.
>
> [Bergemann et al. 2018]: Bergemann, D., Bonatti, A., and Smolin, A. The Design and Price of Information. American Economic Review, 2018.
>
> [Babaioff et al. 2012]: Babaioff, M., Kleinberg, R., and Paes Leme, R. Optimal Mechanisms for Selling Information. In EC 2012

---

> > ### Author Rebuttal · Reviewer_bsm5 · 2026-03-31
> >
> > The rebuttal adds empirical evidence across 4 new settings - 3 real-world FLamby datasets with natural label skew and 1 large market with synthetic market parameters - all of which demonstrate that best-response dynamic converges in heterogeneous buyer settings.
> >
> > The rebuttal also introduces several new settings that vary the parameters meaningfully. The new experiments with substitutable datasets is notable: after 10000 deviations, the best-response dynamics still does not reach an equilibrium i 19% of instances, which itself worth a discussion.
> >
> > Overall, my main concerns are well addressed. I acknowledge and appreciate the meaningful new experiments.

---

> > > ### Author Response · Authors · 2026-04-01
> > >
> > > Thank you very much for your positive feedback and for taking the time to consider our rebuttal. We’re glad that our clarifications and additional experiments helped address your concerns.

---

### Official Review · Reviewer_NiNp · 2026-03-13

**Soundness:** 3
**Presentation:** 4
**Significance:** 3
**Originality:** 4
**Overall Recommendation:** 4
**Confidence:** 4

**Summary:**

This paper proposes a market equilibrium model for data market where data are non-rivalrous and freely replicable. Each seller has one unit of dataset to sell, setting a unit price for fractional allocation. Each buyer has a budget and purchases data bundle to maximize her utility. The paper studies settings where buyers have Leontief (complementary) or linear (substitute) utility, with a common valuation function and asymmetric budgets and utility thresholds. Assuming each buyer always buys her optimal bundle, the market equilibrium is defined as the princing under Nash equilibrium among sellers maximizing their revenue. The paper show that for symmetric Leontief valuation, an equilibrium exists and a proposed algorithm can be compute in polynomial time the equilibrium that maximizes buyer welfare and total seller revenue, with equitable seller revenues. Best response dynamics is proved to converge to $(1+\epsilon)$-approximate equilibrium in polynomial time, but empirical study shows that the resulting equilibrium can be inefficient. Under linear utility of buyers, the equilibrium may not exist when there are multiple buyers with symmetric valuations, but always exists when there is only a single buyer.

**Compliance With Llm Reviewing Policy:**

Affirmed.

**Final Justification:**

My concerns are well addressed by the authors' rebuttal. I am generaly positive about this paper, but I will retain my rating for the submitted version, given the need for revisions and the acknowledged limitations.

**Key Questions For Authors:**

1. In the proof of Lemma A.3, starting from line 795, the inequalities assumed for contradiction should be $\geq$ rather than $>$. This leads to $(K-J)(I-J)\geq 0$ at line 818, which may hold if $K=J$ or $I=J$. Will this affect the soundness of the proof?

**Limitations:**

yes

**Strengths And Weaknesses:**

Strengths:
1. This paper proposes a market equilibrium model for data market, which naturally generalizes classical market equilibrium and captures non-rivalry of data. The proposed equilibrium notion is novel and of practical relevance. The theoretical results are complete, nnd the best-response dynamics is evaluated with empirical experiments, providing practical implications.
2. Techniques for the main results (especially existence for symmetric Leontief valuation and BRD convergence) are non-trivial and differ from standard methods.
3. The main body is well-written with clear proof sketches.

Weaknesses:
1. While the proposed equilibrium notion applies to general valuations, the main results are restricted to symmetric buyer valuations, and only Leontief and linear utilities are considered. This limits the significance of this paper's results, since asymmetric valuations and more general utilitie structures like CES frequently appear in literature and in practice.
2. The empirical evaluation for BRD is conducted in only one synthetic setting, which appears somewhat limited. The meaning of the metrics in table 1 are not explained sufficiently.
3. The appendix would benefit from further refinement to improve readability and help the verification of soundness of proofs.

---

> ### Author Rebuttal · Authors · 2026-03-31
>
> We thank the reviewer for their insightful comments and for appreciating our techniques, novelty, and practical relevance. Below we will address the question and concerns raised by them.
>
> **Q1**: We thank the reviewer for catching this. The inequalities from line 795 should indeed be $\ge$, so line 818 becomes $(K-J)(I-J)\ge 0.$ We will correct this in the revision. This does not affect the proof’s validity: the non-contradictory cases are only the degenerate cases $K=J$ or $I=J$, which arise when multiple buyers share the same threshold $b_i/\tau_i$. Such buyers always become active/inactive simultaneously, so they correspond to the same state rather than distinct states. Thus, without loss of generality, we can merge duplicate threshold states and assume all states have distinct threshold values. Under this assumption, for any three distinct states $i>j>k$, we obtain the strict relation needed for $(K-J)(I-J)<0$, and the contradiction goes through unchanged. Therefore, the reviewer’s correction is valid, but it only reveals a degenerate equality case that is eliminated once duplicate threshold states are identified and merged. The proof remains correct after this clarification. We again thank the reviewer for pointing this out and we will add this clarification in the final version.
>
> **W1**:We agree that extending the model to asymmetric buyers and broader utility classes such as CES is an important direction for future work, and we view our results as a baseline toward that direction.
> Linear and Leontief utilities are canonical benchmark classes in market-equilibrium theory, where the computational perspective of both has been extensively studied in their own right. Likewise, symmetry is a natural first benchmark in several data-market settings where buyers have the same downstream tasks, and even under this assumption our model already goes beyond much of the prior literature, which often considers only a single buyer [Bhawalkar et al. 2025] or even a single buyer–seller interaction [Bergemann et al. 2018, Babaioff et al. 2012]. In contrast, we study a full strategic market with multiple strategic sellers, where standard equilibrium-existence arguments do not directly apply. We nevertheless agree that heterogeneous buyers are an important next step; indeed, in extensive real-world and synthetic experiments with heterogeneous buyers (details at https://rebuttal.tiiny.site), we give empirical evidence of the existence of equilibrium and convergence of best response dynamics (BRD) in markets with heterogeneous buyers under Leontief valuations.
>
> **W2**:
> We thank the reviewer for this comment. Following this suggestion, we substantially broadened the empirical evaluation to additional real-world and synthetic settings covering both complementary and substitutable data; full details are provided at https://rebuttal.tiiny.site. For complementary data with heterogeneous buyers, using real healthcare datasets from FLamby as well as synthetic markets with 100 buyers and sellers, BRD always converges quickly (within O(#sellers) many steps) to a Nash equilibrium, though the equilibrium reached can vary substantially across initial prices in both seller revenue and buyer welfare, highlighting the value of a mediating platform that selects the revenue- or welfare-maximizing equilibrium. For substitutable data, in synthetic markets with one buyer and 100 sellers, behavior is qualitatively different: while complementary instances converge in seconds, similarly sized linear instances fail to converge in over 19% of cases even with 100× more time. When BRD does converge, total seller revenue equals the buyer’s budget and buyer welfare is always at least 0.45 of the optimal equilibrium welfare, suggesting that the main advantage of a mediating platform in these settings is efficient computation of the optimal equilibrium, whereas the BRD can be slow or fail to converge.
> Overall, these additional experiments considerably strengthen our main result: we give empirical evidence of the existence of equilibrium and convergence of BRD in markets with heterogeneous buyers under Leontief valuations.
>
> We agree that Table1 should be explained more clearly and will revise it accordingly. For each random instance, we enumerate all ConstProd-price Nash equilibria and compute the minimum and maximum buyer welfare and total seller revenue attained across any equilibria, denoted by $min W, max W, min R, max R$. We then run BRD from zero prices to obtain the BR equilibrium. Table1 reports the welfare and revenue of this BR equilibrium relative to the best equilibrium values, namely $\mathrm{BR}/\max W$ and $\mathrm{BR}/\max R$, and summarizes these ratios over 100 random instances. Small ratios highlight the value of a mediating platform that computes an optimal equilibrium rather than relying on BRD.
>
> **W3**: Thank you for this helpful feedback. We will revise the appendix to improve its organization, readability, and overall flow.

---

> > ### Author Rebuttal · Reviewer_NiNp · 2026-04-02
> >
> > Thank you for the detailed response. My concerns are well addressed. I am generaly positive about this paper, but I will retain my rating for the submitted version, given the need for revisions and the acknowledged limitations.

---

> > > ### Author Response · Authors · 2026-04-02
> > >
> > > Thank you very much for your careful consideration and for the positive feedback. We understand your decision about the rating. We are glad that the concerns have been addressed and that they can be incorporated as additional clarifications and experiments in the final version. We appreciate your support for the paper’s contributions.

---

### Official Review · Reviewer_wjZ6 · 2026-03-15

**Soundness:** 4
**Presentation:** 4
**Significance:** 3
**Originality:** 4
**Overall Recommendation:** 5
**Confidence:** 3

**Summary:**

The paper studies a data market model where m sellers have items with unlimited supply to sell to n buyers. Each buyer has a fixed budget and a given utility function over any fractional set of m items provided by the sellers, and the sellers decide the prices of each item. In a market equilibrium, each buyer purchases the utility-maximizing bundle (that exceeds their minimum utility requirement), and each seller wants to maximize the revenue from his item.

The paper proves that for homogeneous buyers:
- For Leontief utility functions, the equilibrium always exists, and the social welfare optimal equilibrium is efficiently computable. Furthermore, the best response dynamics among buyers and sellers converges to an equilibrium (but may not be the optimal one).
- For Linear utility functions, the equilibrium may not exist even for two sellers.

**Compliance With Llm Reviewing Policy:**

Affirmed.

**Final Justification:**

I would keep my positive score given that the authors have addressed my concerns.

**Key Questions For Authors:**

Let's ignore any practical justification. Can you elaborate what happens when the minimal utilities are removed? Is this assumption mathematically necessary for the market where the seller does not have production cost?

**Limitations:**

Yes

**Strengths And Weaknesses:**

I don’t have comments on the soundness and originality, as I would rate them positively.

Presentation: The paper is generally well-written, but should definitely discuss the relationship to the classic Fisher Market on what is the same (from buyer side) and what is not (from seller side). The discussion about Leontief and Linear functions immediately reminds me about the literature of Arrow-Debreu and Fisher markets.

(I find it cute to have a result where Leontief markets are solvable and Linear markets are hard, which looks completely different from what happens to Arrow-Debreu markets.)

Significance: The paper studies a very important problem with a good model, and I think the model itself is significant enough to get it accepted. The result and proof become much weaker due to the assumption of homogeneous buyers, though.

---

> ### Author Rebuttal · Authors · 2026-03-31
>
> We thank the reviewer for appreciating our results and model. Below, we will respond to their concerns and questions.
>
> **Fisher Markets vs. Data Markets**:
> Thank you for raising this point. We will add the following comparison. The basic setup for both the markets is the same: They consist of a set of buyers, each with a budget and preferences defined by utility functions, and a set of sellers, each with a unit amount of possibly different goods to sell. Given prices of goods, buyers demand their optimal bundle, that is, a bundle of goods that maximizes their utility subject to their budget constraint.
>
> Notably, in the Fisher market, the allocation rule requires that the total allocation of goods should be at most one (assuming normalization), meaning the same item cannot be allocated to multiple buyers. However, in data markets, the same piece of data can be sold to multiple buyers, and therefore, the main difference between traditional and data markets is that the allocation of data is not bounded by one unit. Concisely, in a data market, there is no allocation constraint on the side of sellers.
>
>
> **Can you elaborate what happens when the minimal utilities are removed? Is this assumption mathematically necessary for the market where the seller does not have production cost?**
> We thank the reviewer for this question. The minimum-utility assumption is mathematically important for the existence of a finite equilibrium, and its role is independent of whether sellers have production costs.
> If the minimum-utility assumption is removed for all buyers, then this corresponds to setting $\tau_i = 0$ for every buyer $i$. In that case, the thresholds $b_i/\tau_i$ are not well defined, and Claim 3.1 implies that, for any finite price vector, each seller can strictly increase its revenue by raising its price further. Hence, no finite Nash equilibrium exists. If the platform instead enforces an exogenous price cap $P$, then the unique equilibrium is the one in which every seller sets price $P$.
> If the assumption is removed only for a subset of buyers, the situation is more delicate. As established in Theorem 3.7, either a Nash equilibrium still exists—when the total budget of the zero-threshold buyers is sufficiently small—or no finite Nash equilibrium exists. Moreover, the correct case can be decided in polynomial time.
> Importantly, this assumption is not tied to sellers’ production costs. It is a buyer-side assumption: in many data markets, a buyer needs to acquire a statistically meaningful amount of data before deriving any benefit, which naturally induces a positive minimum utility threshold. In other words, buyers effectively require a minimum utility level before entering the market.
> Finally, allowing sellers to have a price floor induced by production costs does not resolve the issue above. Claim 3.1 still implies that when all buyers have $\tau_i = 0$, sellers strictly prefer higher prices, so a finite Nash equilibrium still does not exist unless the platform imposes a price cap $P$.

---

> > ### Author Rebuttal · Reviewer_wjZ6 · 2026-04-01
> >
> > My concerns have been adequately addressed.

---

> > > ### Author Response · Authors · 2026-04-02
> > >
> > > Thank you very much for taking the time to consider our rebuttal. We’re glad that our clarifications and additional experiments helped address your concerns.

---

### Decision · Program_Chairs · 2026-04-30

**Decision:**

Accept (regular)

**Comment:**

The paper studies market equilibria in bilateral data markets. All reviewers like the paper, identifying strengths such as the significance of the model and the meaningful and "cute" results. There were minor concerns, most of which were resolved after the author response phase. Overall I believe the paper would be a solid contribution to ICML.